# Molecular rationale for antibody-mediated targeting of the hantavirus fusion glycoprotein

Ilona Rissanen[1,2,3†]*, Robert Stass[1†], Stefanie A Krumm[4], Jeffrey Seow[4], Ruben JG Hulswit[1], Guido C Paesen[1], Jussi Hepojoki[5,6], Olli Vapalahti[7], Åke Lundkvist[8], Olivier Reynard[9], Viktor Volchkov[9], Katie J Doores[4], Juha T Huiskonen[1,2,3], Thomas A Bowden[1]*

[1]Division of Structural Biology, Wellcome Centre for Human Genetics, University of Oxford, Oxford, United Kingdom; [2]Helsinki Institute of Life Science HiLIFE, University of Helsinki, Helsinki, Finland; [3]Molecular and Integrative Biosciences Research Programme, The Faculty of Biological and Environmental Sciences, University of Helsinki, Helsinki, Finland; [4]Department of Infectious Diseases, King's College London, Guy's Hospital, London, United Kingdom; [5]Institute of Veterinary Pathology, Vetsuisse Faculty, University of Zürich, Zürich, Switzerland; [6]Department of Virology, Medicum, Faculty of Medicine, University of Helsinki, Helsinki, Finland; [7]Departments of Virology and Veterinary Biosciences, University of Helsinki and HUSLAB, Helsinki University Hospital, Helsinki, Finland; [8]Zoonosis Science Center, Department of Medical Biochemistry and Microbiology, Uppsala University, Uppsala, Sweden; [9]CIRI, Centre International de Recherche en Infectiologie, INSERM U1111, CNRS UMR5308, Université Lyon, Lyon, France

*For correspondence:
ilona.rissanen@helsinki.fi (IR);
thomas.bowden@strubi.ox.ac.uk
(TAB)

†These authors contributed
equally to this work

Competing interests: The
authors declare that no
competing interests exist.

Reviewing editor: Pamela J
Bjorkman, California Institute of
Technology, United States

**Abstract** The intricate lattice of Gn and Gc glycoprotein spike complexes on the hantavirus envelope facilitates host-cell entry and is the primary target of the neutralizing antibody-mediated immune response. Through study of a neutralizing monoclonal antibody termed mAb P-4G2, which neutralizes the zoonotic pathogen Puumala virus (PUUV), we provide a molecular-level basis for antibody-mediated targeting of the hantaviral glycoprotein lattice. Crystallographic analysis demonstrates that P-4G2 binds to a multi-domain site on PUUV Gc and may preclude fusogenic rearrangements of the glycoprotein that are required for host-cell entry. Furthermore, cryo-electron microscopy of PUUV-like particles in the presence of P-4G2 reveals a lattice-independent configuration of the Gc, demonstrating that P-4G2 perturbs the $(Gn-Gc)_4$ lattice. This work provides a structure-based blueprint for rationalizing antibody-mediated targeting of hantaviruses.

## Introduction

Rodent-borne hantaviruses (genus *Orthohantavirus*, family *Hantaviridae*) are enveloped, negative-sense RNA viruses found worldwide in small mammals (*Jonsson et al., 2010*; *Watson et al., 2014*). Cross-species transmission into humans typically results from inhalation of aerosolized excreta from chronically infected rodents and has two primary clinical outcomes. Hantaviral species native to Europe and Asia cause hemorrhagic fever with renal syndrome (HFRS), while those in the Americas cause hantavirus cardiopulmonary syndrome (HCPS) (*Lee and van der Groen, 1989*; *Watson et al., 2014*; *Zaki et al., 1995*). The case mortality rate may exceed 30% for HCPS and ranges from <1% to 15% for HFRS (*Alonso et al., 2019*; *Heyman et al., 2011*; *Jonsson et al., 2010*). Despite variable clinical characteristics, both syndromes arise from excessive proinflammatory and cellular immune

responses to infection (*Maes et al., 2004*; *Terajima and Ennis, 2011*). Due to their potential to cause severe disease and to be transmitted by aerosols, hantaviruses have been identified as potential bioterrorism agents by the Centers for Disease Control and Prevention (*Moran, 2002*).

Entry of a hantavirus into a host cell is negotiated by two membrane-anchored glycoproteins, Gn and Gc, which form a lattice decorating the lipid bilayer envelope of the mature virion (*Battisti et al., 2011*; *Hepojoki et al., 2010*; *Huiskonen et al., 2010*). Gn and Gc are translated as a single polypeptide and cleaved at a 'WAASA' signal sequence during protein folding and assembly (*Löber et al., 2001*; *Schmaljohn et al., 1987*). Recent studies have reported crystal structures of both Gn (*Li et al., 2016*; *Rissanen et al., 2017*) and Gc (*Guardado-Calvo et al., 2016*; *Willensky et al., 2016*) ectodomains, and their higher-order, tetrameric organization has been postulated based on biochemical characterization (*Hepojoki et al., 2010*) and cryo-electron microscopy reconstructions of purified virions (*Battisti et al., 2011*; *Huiskonen et al., 2010*; *Li et al., 2016*). The N-terminal region of the Gn ectodomain exhibits a mixed α/β fold, which locates to the membrane-distal region of the tetrameric spike and is linked to the virion envelope by a C-terminal stalk region (*Huiskonen et al., 2010*; *Li et al., 2016*; *Rissanen et al., 2017*). The Gc adopts a three-domain (domains I−III) class II fusion protein architecture and is postulated to associate closely with the Gn, linking adjacent Gn tetramers (*Guardado-Calvo et al., 2016*; *Hepojoki et al., 2010*; *Li et al., 2016*; *Willensky et al., 2016*). The Gn has been shown to shield the fusion loop resident in domain II of the Gc, in a manner similar to that proposed in phleboviruses and alphaviruses (*Allen et al., 2018*; *Guardado-Calvo et al., 2016*; *Guardado-Calvo and Rey, 2017*; *Halldorsson et al., 2018*; *Li et al., 2016*; *Serris et al., 2020*; *Voss et al., 2010*). Following host cell binding and endocytotic uptake of the virus, acidification in endosomal compartments results in at least partial dissolution of the Gn−Gc lattice (*Acuña et al., 2015*; *Hepojoki et al., 2010*; *Rissanen et al., 2017*). Endosomal escape is mediated by the Gc in a process that involves engagement of the target membrane by the Gc-encoded fusion loop, followed by upward 'zippering action' conformational rearrangements of the transmembrane-anchored Gc domain III, which bring together host and viral membranes (*Guardado-Calvo et al., 2016*; *Willensky et al., 2016*).

In contrast to the chronic, asymptomatic infection observed in the native rodent reservoir of a given hantaviral species, human infections are often acute and characterized by a strong immune response (*Schönrich et al., 2008*). Indeed, the generation of robust antibody-mediated immune response against Gn and Gc has been shown to be key in limiting disease progression in humans, where high titers of neutralizing antibodies (nAbs) in the acute phase correlate with a more benign course of disease (*Bharadwaj et al., 2000*; *Dantas et al., 1986*; *Pettersson et al., 2014*). Furthermore, a neutralizing humoral immune response also conveys long-lasting immunity against human hantaviral infection (*Lundkvist et al., 1993b*; *Schmaljohn et al., 1990*; *Settergren et al., 1991*).

Although elicitation of a productive nAb response constitutes an essential component of the host immune response to hantavirus infection (*Arikawa et al., 1989*; *Heiskanen et al., 1999*; *Levanov et al., 2019*; *Lundkvist and Niklasson, 1992*), there is a paucity of information with regard to the molecular determinants of antibody-mediated neutralization. Here we structurally characterize the epitope of mAb P-4G2, a potently neutralizing antibody derived following experimental infection of bank voles with Puumala virus (PUUV), a hantavirus responsible for one of the most frequently occurring hantaviral diseases, a mild form of HFRS known as *nephropathia epidemica* (NE) (*Heyman et al., 2011*; *Linderholm and Elgh, 2001*). Through a combined crystallographic and electron cryo-tomography (cryo-ET) analysis, we report the structure of mAb P-4G2 in complex with PUUV Gc and define the P-4G2 epitope on the Gc subcomponent of the mature Gn−Gc spike. This work provides the first molecular-level insights into antibody-mediated targeting of the antigenic hantaviral surface.

## Results

### Recombinantly derived and Gc-specific bank vole mAb P-4G2 potently neutralizes PUUV

The hybridoma cell line producing the Gc-specific neutralizing mAb P-4G2 was reported in 1992 following experimental infection of the natural reservoir species, bank voles (*Myodes glareolus*), with PUUV (*Lundkvist and Niklasson, 1992*). While mAb P-4G2 has been shown to be potently

neutralizing against PUUV, with a mAb concentration of ~4 nM (0.6 µg/ml) (*Lundkvist and Niklasson, 1992*) resulting in 80% inhibition of PUUV infection in focus reduction neutralization tests (*Heiskanen et al., 1997*; *Lundkvist and Niklasson, 1992*; *Lundkvist et al., 1993c*), there has been a paucity of information with regard to the structural basis for neutralization by this or any other Gc-specific antibody. We rescued the sequence of the P-4G2 variable heavy ($V_H$) and kappa ($V_K$) chain regions (see *Figure 1—figure supplement 1*) from hybridoma cells by PCR with gene-specific primers originally designed for mouse $V_H$ and $V_K$ chains (*von Boehmer et al., 2016*). The resulting recombinantly produced mAb P-4G2 comprised bank vole variable and mouse constant regions and, consistent with previously reported in vitro studies of hybridoma-derived mAb P-4G2 (*Hepojoki et al., 2010*; *Lundkvist and Niklasson, 1992*), potently neutralizes PUUV in a hantavirus-pseudotyped VSV-ΔG RFP (Red Fluorescent Protein) neutralization assay at an $IC_{50}$ value of 0.088 µg/ml (*Figure 1*).

## Crystal structure of the Fab P-4G2 in complex with PUUV Gc reveals the P-4G2 epitope on pre-fusion Gc

To ascertain the molecular basis for P-4G2-mediated recognition of PUUV, we determined the crystal structure of the PUUV Gc ectodomain in complex with the antigen-binding fragment (Fab) of P-4G2 to 3.50 Å resolution (*Figure 2*, *Figure 2—figure supplement 1*, and *Supplementary file 1*). Consistent with previous studies of hantaviral Gc glycoproteins (*Guardado-Calvo et al., 2016*; *Willensky et al., 2016*), PUUV Gc assumes a class II fusion glycoprotein architecture composed of three domains (domains I−III), with domain I and domain II forming an elongated structure and domain III forming inter-domain contacts at the side of domain I. Fab P-4G2 recognizes a protein-specific epitope at the junction of domain I and domain II of PUUV Gc, distal from the hydrophobic fusion loop at the tip of domain II (*Figure 2*). The epitope comprises ~830 Å² of buried surface area and all complementarity-determining region (CDR) loops participate in the interaction (*Figure 2—figure supplement 2*). Arg100 (CDRH3) is the most centrally located paratope residue, and is predicted to form multiple hydrogen bonds and a putative salt bridge with Gc residue Glu725 and across the antibody−antigen interface (*Figure 2B* and *Figure 2—figure supplement 2*; interacting residues were identified using the PDBePISA server [*Krissinel and Henrick, 2007*]). Site-directed mutagenesis of Arg100 to Ala considerably reduced the potency of P-4G2 to neutralize PUUV

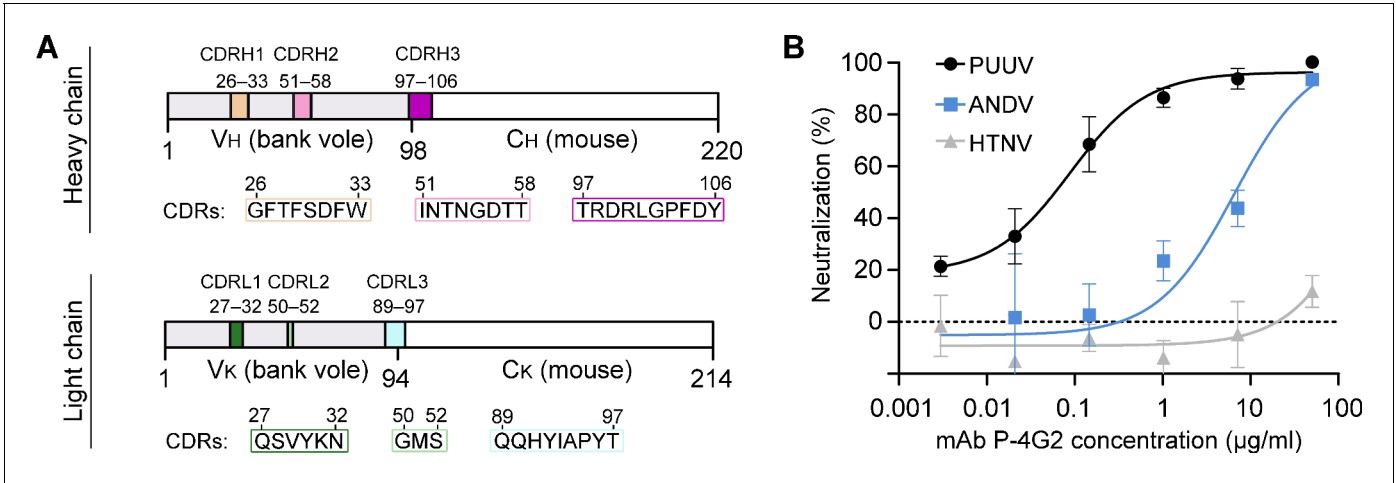

**Figure 1.** Composition and neutralization potency of recombinantly-derived bank vole mAb P-4G2. (**A**) Composition of the complementarity-determining regions (CDRs) of the mAb P-4G2 antigen-binding fragment (Fab) heavy ($V_H$) and kappa ($V_K$) chains. For the full sequence of Fab P-4G2 variable regions, please see *Figure 1—figure supplement 1*. (**B**) A hantavirus-pseudotyped VSV-ΔG RFP neutralization assay shows that recombinantly produced mAb P-4G2 neutralizes Puumala virus- and Andes virus-pseudotyped VSV (black and blue traces, respectively), but not Hantaan virus-pseudotyped VSV (gray trace). Each neutralization assay was carried out three times in duplicate. A representative experiment is shown. Error bars represent the range of the value for the experiment performed in duplicate.

The online version of this article includes the following figure supplement(s) for figure 1:

**Figure supplement 1.** Sequence alignment of antibody variable regions from bank vole mAb P-4G2 and a representative mouse antibody.

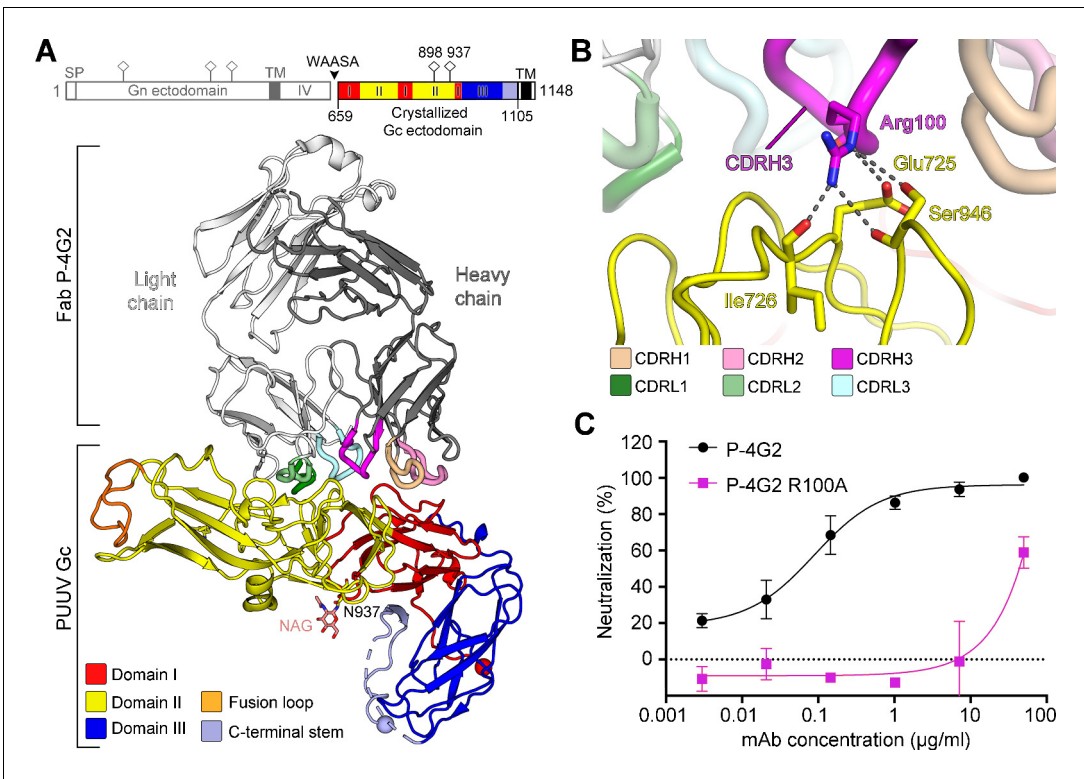

**Figure 2.** Crystal structure of neutralizing antibody P-4G2 in complex with Puumala virus (PUUV) Gc. (**A**) Crystal structure of Fab P-4G2–PUUV Gc complex at 3.5 Å resolution. PUUV Gc, a class II fusion protein, comprises domains I–III (colored red, yellow, and blue, respectively), a Gc C-terminal tail (light blue) and the viral fusion loop at Ser771–Thr785 (orange). Fab P-4G2, comprised of a heavy chain (dark gray) and a light chain (white), is observed bound at the junction of domains I and II on PUUV Gc. A domain schematic of the PUUV glycoprotein precursor with the signal peptide (SP), transmembrane domains (TM), intra-virion domain (IV), and WAASA signal peptidase cleavage site is shown alongside the construct used in crystallization (schematic was produced using the DOG software [***Ren et al., 2009***]). N-linked glycosylation sequons are shown in the domain schematic as pins and the N-linked glycan observed at Asn937 is rendered as sticks. There was no evidence of glycosylation at Asn898. (**B**) A close-up view of the central portion of the Fab P-4G2-PUUV Gc interface around paratope residue Arg100 of the Fab P-4G2 CDRH3. Residues contributing to a putative hydrogen bonding network (indicated by dashed lines) are rendered as sticks and were identified using the PDBePISA server (***Krissinel and Henrick, 2007***). (**C**) Site-directed mutagenesis of Arg100 to Ala reduces the neutralizing potency of mAb P-4G2, confirming the importance of this residue in the interaction interface. We compared the neutralization potency of mAb P-4G2 R100A mutant (magenta trace) to that of the wild-type mAb P-4G2 (black trace) against PUUV pseudovirus in a hantavirus-pseudotyped VSV-ΔG RFP neutralization assay. The virus neutralization assay presented in panel C was conducted as a part of the same experiment as the neutralization assay presented in ***Figure 1B***. Each neutralization assay was carried out three times in duplicate. A representative experiment is shown. Error bars represent the range of the value for the experiment performed in duplicate.

The online version of this article includes the following figure supplement(s) for figure 2:

**Figure supplement 1.** Electron density at the Fab P-4G2–Puumala virus (PUUV) Gc interface.

**Figure supplement 2.** Key interactions at the Fab P-4G2–Puumala virus Gc complex interface.

**Figure supplement 3.** Sequence conservation at the mAb P-4G2 epitope.

(***Figure 2C***). One Fab P-4G2–PUUV Gc complex was observed in the crystallographic asymmetric unit, and there was no evidence for the formation of higher-order Gc dimers or trimers, such as those observed in pre- and post-fusion class II fusion glycoprotein structures (***Modis, 2013***).

Consistent with the hypothesis that mAb P-4G2 targets a pre-fusion conformation of PUUV Gc representative of that displayed on the native virion surface, structure overlay analysis reveals that PUUV Gc from the Gc–P-4G2 complex (PUUV Gc$_{Gc–P-4G2}$) is highly similar to the recently reported pre-fusion state of Andes virus (ANDV) Gc (***Serris et al., 2020***), and is distinct from the known PUUV

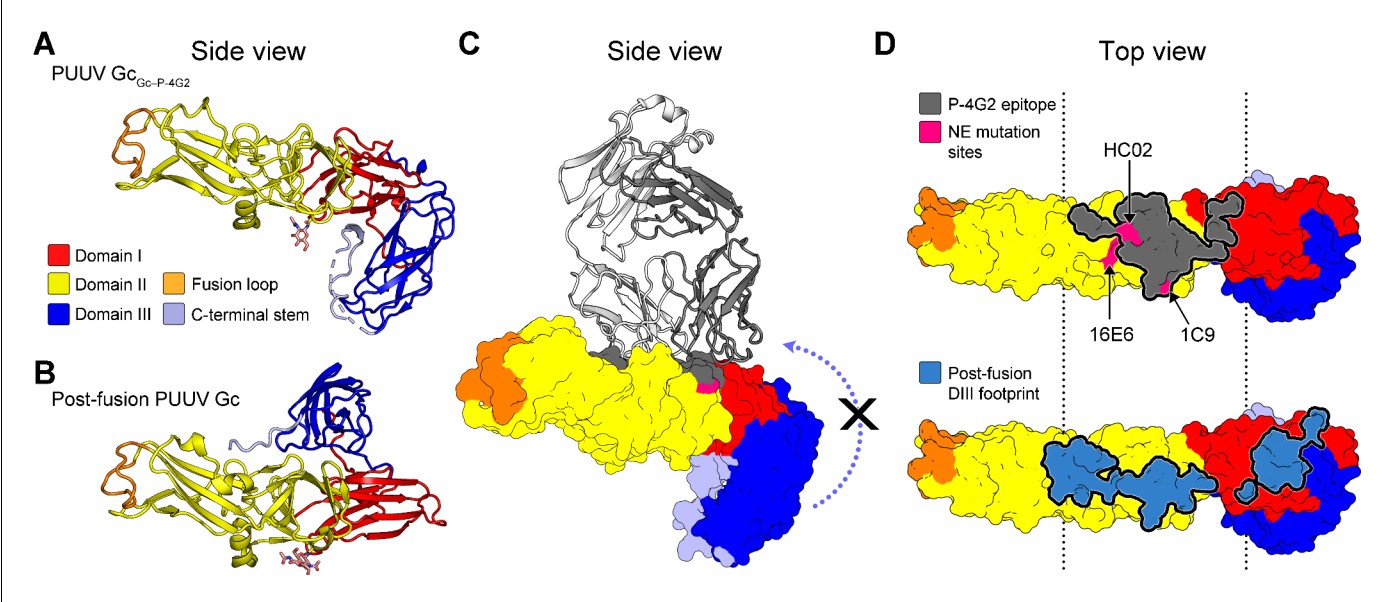

**Figure 3.** The epitope of antibody P-4G2 denotes a key antigenic site at the hantaviral surface. (**A**) Puumala virus (PUUV) Gc from the Gc-4G2 complex displays a domain III conformation distinct from that observed in (**B**) post-fusion PUUV Gc (*Willensky et al., 2016*). (**C**) Fab P-4G2 binding precludes the rearrangement of domain III (blue arrow) to the post-fusion conformation. PUUV Gc is represented as a surface, with the epitope of Fab P-4G2 outlined dark gray. Fab P-4G2 is shown in ribbon representation, with heavy and light chains colored gray and white, respectively. (**D**) Top view of the P-4G2 epitope shows that it overlaps with the neutralization evasion mutation sites reported for PUUV-neutralizing human antibody 1C9 (*Hörling and Lundkvist, 1997*; *Lundkvist et al., 1993a*) and HTNV-neutralizing mouse antibodies HC02 and 16E6 (*Arikawa et al., 1989*; *Wang et al., 1993*). Furthermore, the epitope overlaps with the binding site of domain III in the post-fusion conformation (medium blue). Interfacing residues were identified using the PDBePISA server (*Krissinel and Henrick, 2007*).

The online version of this article includes the following figure supplement(s) for figure 3:

**Figure supplement 1.** Overlay analysis of viral class II fusion proteins with Puumala virus (PUUV) Gc$_{Gc-P-4G2}$ indicates that PUUV Gc has crystallized in a conformation similar to the pre-fusion conformation.

Gc post-fusion structure (*Willensky et al., 2016*; *Figure 3* and *Figure 3—figure supplement 1*). Notably, the conformations of domain III from both the recently determined ANDV Gc (*Serris et al., 2020*) and our PUUV Gc contrast those observed across hantavirus Gc structures (*Figure 3—figure supplement 1*), suggesting that this region is likely flexible in the absence of the stabilizing environment of the glycoprotein lattice or crystal environments. Additionally, the conformation of PUUV Gc$_{Gc-P-4G2}$ closely resembles class II fusion protein pre-fusion states, such as those observed for RVFV Gc (PDB 4HJ1) and ZIKV E (PDB 5IRE) (*Figure 3—figure supplement 1*).

Investigation into the residues that comprise the P-4G2 epitope revealed that many of these residues are buried in the post-fusion trimeric Gc (*Figure 3D*), indicating that mAb P-4G2 is specific to the pre-fusion state of PUUV Gc. This observation is consistent with previous experimental findings, which demonstrated that mAb P-4G2 did not recognize PUUV under low pH conditions but remained bound to the virus and protected the epitope if introduced prior to low-pH exposure (*Hepojoki et al., 2010*). Taken together, these results support a model whereby P-4G2 specifically targets pre-fusion Gc and may sterically preclude the formation of a fusogenic configuration of the Gc.

## The epitope of P-4G2 is targeted for neutralization across hantaviral species

Hantaviruses are often divided into two groups, termed Old World and New World hantaviruses (*Jonsson et al., 2010*), reflecting their distribution and pathobiological features. Interestingly, while PUUV is an Old World hantavirus prevalent in Northeastern Europe, the residues comprising the P-4G2 epitope on PUUV Gc exhibit a relatively high level of sequence conservation with Gc proteins

from New World hantaviruses, such as ANDV (72% sequence identity across the P-4G2 epitope), and a lower level of conservation with Hantaan virus (HTNV), an Old World hantavirus (38% sequence identity across the epitope) (*Figure 2—figure supplement 3*). The epitope-based sequence analysis provides a rational basis for understanding the cross-reactivity of P-4G2 with ANDV but not HTNV, where we identify a greater level of sequence variation on HTNV Gc at the epitope and fewer sequence differences on ANDV Gc (*Figure 2—figure supplement 3*). The observed similarity at the epitope between PUUV and New World hantaviruses is in line with the evolutionary history of these viruses and their hosts (*Plyusnin and Sironen, 2014*), and provides a structural basis for the ability of P-4G2 to neutralize ANDV, albeit at a lower $IC_{50}$ value (6.48 µg/ml) (*Figure 1B*). Furthermore, despite the sequence variation at the P-4G2 epitope between the two Old World hantaviruses PUUV and HTNV (*Figure 2—figure supplement 3*), and the lack of HTNV cross-neutralization (*Figure 1B*), we note that the observed P-4G2 binding site overlaps with the putatively assigned binding sites of HTNV-neutralizing mAbs HCO2 and 16E6 (*Arikawa et al., 1989*; *Wang et al., 1993*; *Figure 3*), indicating that this region of the Gc is likely to be immunologically accessible across hantaviral species.

## Mab P-4G2 is specific to a lattice-free state of virion-displayed PUUV Gc

Previous investigations of the non-pathogenic viral orthologue, Tula virus (TULV), have revealed the ultrastructure of Gn−Gc assemblies, as displayed on the hantaviral envelope (*Huiskonen et al., 2010*; *Li et al., 2016*). These studies indicate that the Gc forms elongated structures that extend from the virion membrane and are shielded by the cognate Gn (*Li et al., 2016*).

To ascertain the position of the P-4G2 epitope in the context of the hantaviral envelope, we generated PUUV virus-like particles (VLPs) by transient expression of the PUUV genomic M-segment in mammalian cells (*Acuña et al., 2014*). Cryo-ET of purified PUUV VLPs, in tandem with sub-tomogram averaging, revealed that the VLP surface displays the expected Gn−Gc spike architecture that can form locally ordered lattices, as observed in native hantavirions (*Figure 4—figure supplement 1*). Consistent with previous studies of hantavirus ultrastructure and Gc oligomerization (*Bignon et al., 2019*; *Li et al., 2016*), breakpoints exist in the $(Gn−Gc)_4$ lattices (*Huiskonen et al., 2010*; *Li et al., 2016*; *Serris et al., 2020*). Previous studies have suggested that contacts between $(Gn−Gc)_4$ spikes are mediated by Gc homo-dimers (*Bignon et al., 2019*; *Huiskonen et al., 2010*; *Li et al., 2016*), while lattice breaks are expected to display Gc ('lattice-free Gc') molecules that are not integrated within the lattice assembly through Gc homo-dimer interactions.

Following the validation that PUUV VLP presents a glycoprotein architecture that accurately resembles the native virion, we applied an identical cryo-ET approach to VLPs treated with Fab P-4G2. The data were split in two sets: (i) spikes that were part of a lattice, and (ii) spikes in regions of incomplete lattice. The former yielded a reconstruction similar to the VLP sample prepared without the Fab (*Figure 4A* and *Figure 4—figure supplement 1*), while the reconstruction focused on lattice-free spikes yielded a discrete $(Gn−Gc)_4$ spike with additional density (*Figure 4B*). These results suggest that mAb P-4G2 binding is incompatible with glycoprotein lattice formation, and further, that Fab P-4G2 binding may affect the higher order lattice assembly. To test this hypothesis, we quantified the number of neighbors for each spike on VLPs both in the presence and absence of Fab P-4G2. The frequency of lattice-free spikes (with zero neighbors) was higher in the presence of Fab P-4G2, while the frequency of lattice-bound spikes (with four or more neighbors) was higher in the particles not treated with Fab P-4G2 (*Figure 4C and D* and *Figure 4—figure supplement 2*). We note that treatment with Fab P-4G2 had a heterogeneous effect on the lattice of different VLPs, as can be observed in *Figure 4—figure supplement 2*. These results suggest that Fab P-4G2 influences the presentation of the Gn−Gc lattice by dislodging the $(Gn−Gc)_4$ spikes from each other.

Fitting of the crystal structure of Fab P-4G2−PUUV Gc complex and PUUV Gn (*Li et al., 2016*) into the reconstruction of the Fab-treated $(Gn−Gc)_4$ spike confirms that the additional density is comprised of Fab P-4G2 (*Figure 5*). PUUV $Gc_{Gc−P-4G2}$ fits well within the cryo-ET reconstruction, where the crystallographically observed conformation of domain III accurately matches a feature of the glycoprotein spike (*Figure 5C*), further supporting that PUUV Gc has crystallized in the pre-fusion conformation presented on the mature virion. Additionally, this fitting confirms the location of Gc in the density spanning from the membrane-distal globular lobes to the viral membrane, as we and others have proposed previously (*Li et al., 2016*; *Serris et al., 2020*). The fitting locates domain

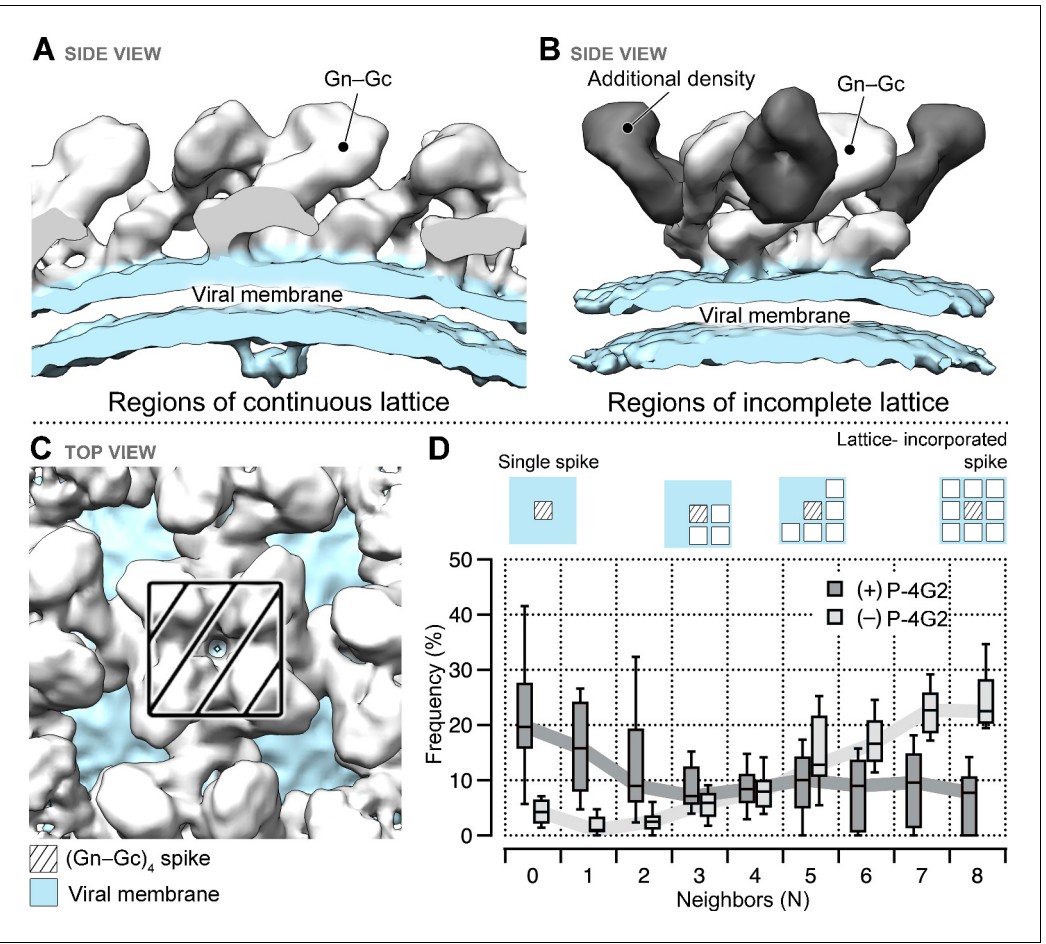

**Figure 4.** Treatment of Puumala virus (PUUV) virus-like particles (VLPs) with Fab P-4G2 results in additional density and is associated with loss of continuous lattice at the VLP surface. Cryo-ET reconstructions of the Fab P-4G2-treated PUUV VLP surface, derived from (**A**) regions of continuous lattice (14.3 Å) and (**B**) regions of incomplete lattice (13.4 Å). While both reconstructions show the canonical Gn−Gc architecture (density colored white) and the viral lipid bilayer (light blue), additional density is observed in the latter reconstruction (dark gray). (**C**) The hantaviral surface carries tetragonal $(Gn-Gc)_4$ spikes that can organize in patches of ordered lattice. (**D**) A box plot describing the frequency of $(Gn-Gc)_4$ spikes that have a given number of lattice compatible neighbors, from zero to a maximum of eight, shows that treatment with Fab P-4G2 alters the presentation of the $(Gn-Gc)_4$ spike assemblies at the VLP surface.

The online version of this article includes the following figure supplement(s) for figure 4:

**Figure supplement 1.** The Puumala virus (PUUV) virus-like particle (VLP) surface displays ordered regions of glycoprotein lattice and is congruent with previously published reconstructions.

**Figure supplement 2.** Treatment with Fab P-4G2 alters the presentation of the hantaviral glycoprotein lattice at the surface of Puumala virus (PUUV) virus-like particles (VLPs).

III of the Gc adjacent to the membrane, allowing the transmembrane region of the C-terminus to be linked to the virion envelope (*Figures 5A and 2A*), and places the fusion loop from Gc domain II into membrane-distal lobe density in close contact with Gn (*Figure 5A and B*), in a manner similar to that observed for other class II fusion proteins and their cognate accessory proteins (*Halldorsson et al., 2018*; *Voss et al., 2010*).

Drawing upon the fitting generated above, we used a 13.9 Å cryo-ET reconstruction of the lattice-incorporated (non-Fab P-4G2 treated) PUUV VLP surface to create a model of the higher-order $(Gn-Gc)_4$ spike lattice. While the overall assembly of $(Gn-Gc)_4$ spikes remain similar to that obtained from fitting into the cryo-ET reconstruction of an individual, Fab-treated glycoprotein spike (*Figure 5*) fitting into the latticed reconstruction provides a structural basis for the formation of

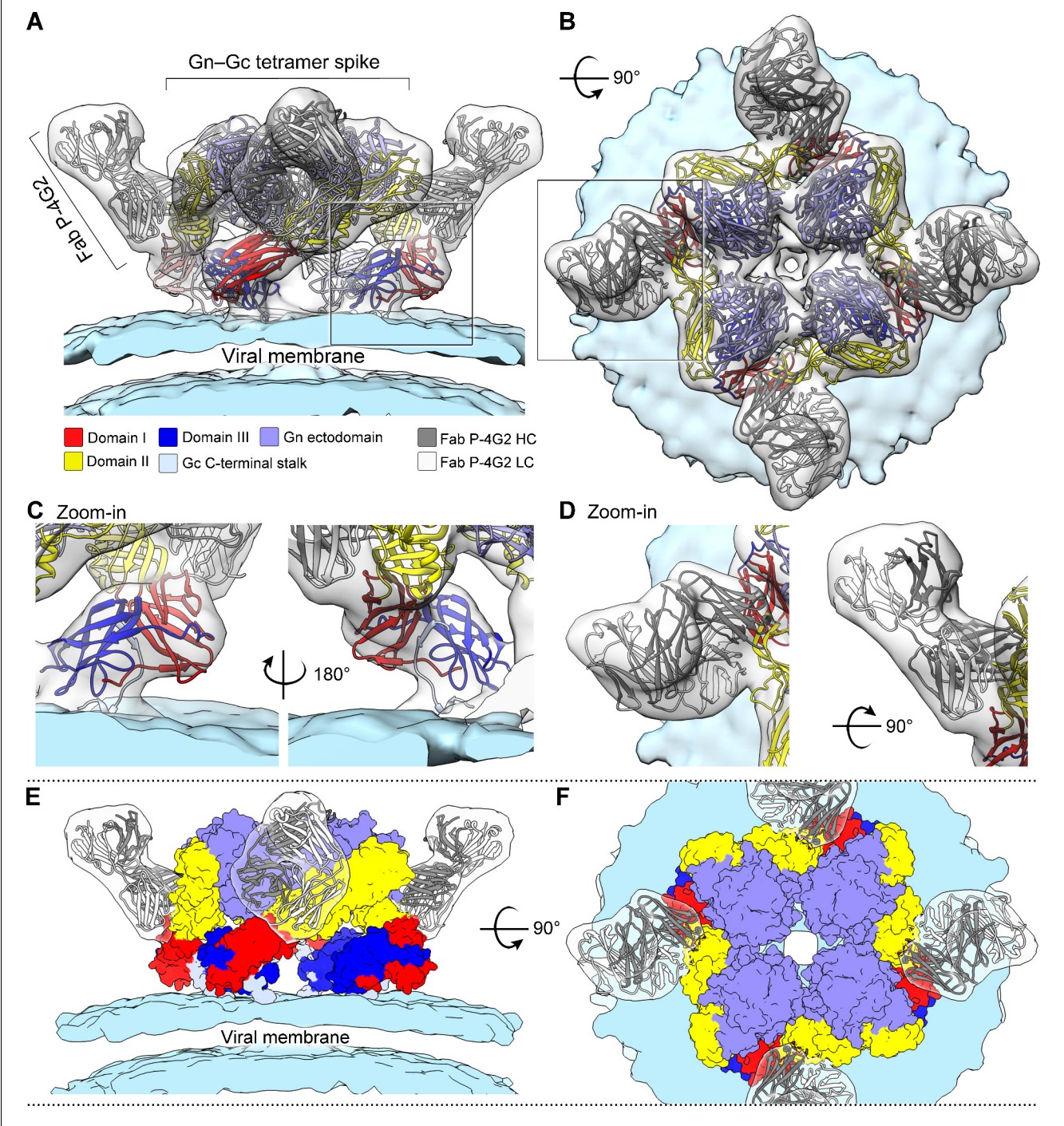

**Figure 5.** Fitting of the Fab P-4G2–Puumala virus (PUUV) Gc crystal structure into the cryo-ET reconstruction confirms the P-4G2 epitope in the context of the viral surface and supports the hypothesis that the observed Gc conformation constitutes a pre-fusion state. (A) Side view and (B) top view of the Fab P-4G2-treated PUUV virus-like particle (VLP) spike at 13.4 Å resolution. Crystal structures of Fab P-4G2–PUUV, along with PUUV Gn (PDB id 5FXU), were fitted in the cryo-ET reconstruction as rigid bodies and display excellent conformity with the cryo-ET derived envelope. (C) A zoom-in of the fit of domain III of Gc into the cryo-ET reconstruction presented from two points of view, which are related by a 180° rotation. The goodness of fit supports the hypothesis that domain III has crystallized in a conformation that closely resembles the pre-fusion state presented in the mature hantaviral spike. (D) A close-up of Fab P-4G2 similarly presents a high-correlation fit. A simplified look at the composition of the hantaviral spike, where fitted Gc and Gn are represented as surfaces, is shown from side view (E) and top view (F).

PUUV Gc homo-dimers at contact points between neighboring spikes (*Figure 6*), which closely resemble a crystallographically observed homo-dimer interface reported previously for HTNV Gc (*Bignon et al., 2019*; *Guardado-Calvo et al., 2016*; *Figure 6—figure supplement 1*). Furthermore, modeling of Fab P-4G2 binding onto the Gc homodimer demonstrates that neighboring epitopes are not mutually accessible for Fab binding in their lattice-integrated form, providing clues to how Fab P-4G2 recognition may be accompanied by dissociation of the Gc homodimer assembly (*Figure 6—figure supplement 2*). Altogether, our combined crystallographic and cryo-EM investigation reveals, in molecular detail, that P-4G2 is specific to the lattice-free, pre-fusion state of PUUV Gc presented on the mature viral envelope.

## Discussion

Despite the importance of neutralizing antibodies in mitigating the progression of hantaviral disease (*Bharadwaj et al., 2000*; *Dantas et al., 1986*; *Pettersson et al., 2014*) and conveying long-lasting protection against infection (*Settergren et al., 1991*; *Valdivieso et al., 2006*; *Ye et al., 2004*), little is known about how they target virion-displayed glycoproteins, Gn and Gc. Here we provide a molecular-level description of the epitope targeted by mAb P-4G2, a host-derived antibody that potently neutralizes PUUV, a zoonotic hantavirus prevalent in Northern Europe and Russia. We find that mAb P-4G2 targets an epitope at the junction of domains I and II of PUUV Gc (*Figure 2*). When combined with a cryo-ET reconstruction of Fab P-4G2 complexed with PUUV (Gn−Gc)₄ spike, our analysis reveals that mAb P-4G2 targets a pre-fusion, lattice-free conformation of PUUV Gc.

Our structural data provides several molecular-level insights into hantavirus Gc functionality and antibody-mediated neutralization. First, comparison of our PUUV Gc pre-fusion state with that of the previously reported PUUV Gc post-fusion state clarifies the conformational changes that PUUV Gc undergoes to facilitate membrane fusion (*Figure 3* and *Figure 3—figure supplement 1*). Our observed PUUV Gc conformation matches that of the Gc in a recently reported ANDV Gn–Gc pre-fusion complex (*Serris et al., 2020*; *Figure 3—figure supplement 1*), further supporting the hypothesis that our PUUV Gc forms a pre-fusion conformation. Interestingly, domain III from a

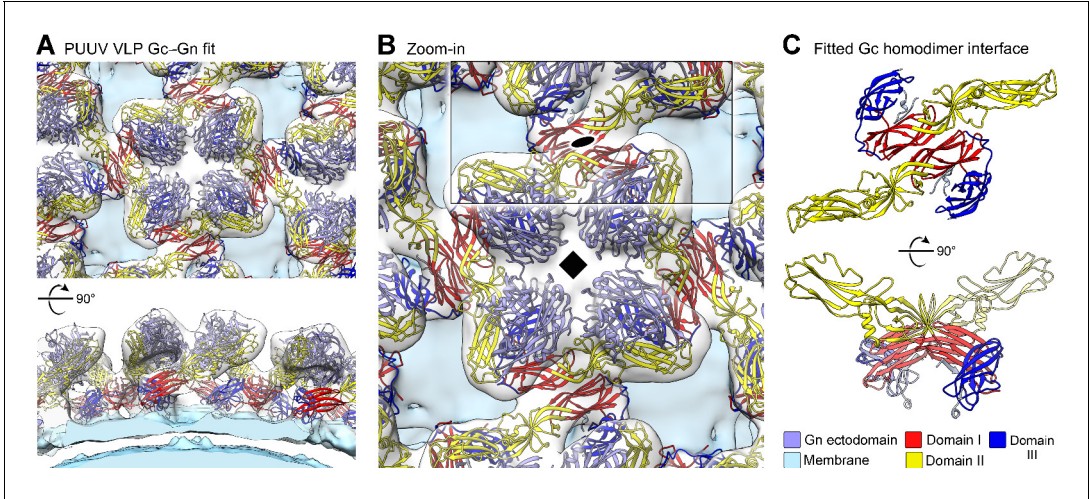

**Figure 6.** Fitting of Puumala virus (PUUV) Gn and Gc into the cryo-ET-derived reconstruction of PUUV virus-like particles (VLPs) shows that lattice formation is mediated by Gc homo-dimers. (A) Top- and side views of the PUUV lattice. Crystal structures of PUUV Gc from the Fab P-4G2 complex, and PUUV Gn (PDB id: 5FXU) are shown fitted into a cryo-ET reconstruction of the PUUV VLP surface at 13.9 Å resolution. PUUV Gn and Gc are shown in cartoon representation and colored as presented at the bottom of panel C. (B) A zoom-in of an individual spike reveals that contacts between spikes are mediated by Gc. (C) The fitting identifies the topology of Gc dimers formed between neighboring spikes. Gc domain I is indicated to be involved in the formation of the dimer interface, corroborating the model reported by Bignon et al. (*Bignon et al., 2019*) based on a crystallographically observed HTNV Gc homo-dimer (*Guardado-Calvo et al., 2016*).

The online version of this article includes the following figure supplement(s) for figure 6:

**Figure supplement 1.** Domain I mediates dimer contacts at the Gc–Gc interface.

**Figure supplement 2.** Fab P-4G2 epitopes from neighboring Puumala virus (PUUV) Gc proteins within the Gn−Gc lattice are in close proximity.

previously reported structure of HTNV Gc adopts a conformation that is intermediate between pre- and post-fusion conformations of PUUV Gc (*Guardado-Calvo et al., 2016*; *Figure 3—figure supplement 1*). Similar structural intermediates have been identified in previous structural studies of related phlebovirus Gc glycoproteins, alone (*Dessau and Modis, 2013*) and in the context of the virion (*Halldorsson et al., 2018*), which is consistent with domain III flexibility being an intrinsic feature required for Gc glycoprotein functionality.

Second, examination of the antibody−antigen interface provides a structure-based model for understanding the mechanism of P-4G2-mediated neutralization, where overlay analysis reveals that Fab P-4G2 binding is incompatible with the pH-dependent conformational transitions of PUUV Gc required to negotiate fusion of the viral and cellular membranes. While it is also possible that mAb P-4G2 interferes with other steps of the host-cell entry pathway (e.g. receptor recognition), our modeling indicates that the antibody targets residues crucial for the formation of a post-fusion Gc configuration (*Figure 3*). Indeed, binding to this site likely impedes the upward-occurring 'zippering action' of domain III during Gc-mediated viral fusion (*Guardado-Calvo et al., 2016*; *Willensky et al., 2016*). This proposed structure-based mechanism of neutralization is supported by previously reported in vitro studies, which demonstrate that mAb P-4G2 protects the epitope upon low-pH exposure (*Hepojoki et al., 2010*). Further support for this hypothesis can be gleaned from previous reports, which have shown that hindering such movements of domain III, via either mutagenesis or the introduction of competing peptides (*Barriga et al., 2016*), drastically reduces viral infectivity. Given that small molecule ligands have also been shown to prevent fusogenic rearrangements on the analogous DI−DII interface region of flavivirus E glycoproteins (*Chao et al., 2018*; *Modis et al., 2003*), which also bear a class II fusion architecture, it will be interesting to assess whether small molecule inhibitor development may be similarly successful against this region on the hantavirus Gc.

In line with the hypothesis that the identified epitope on Gc domains I and II may constitute a site of vulnerability across hantaviruses, more generally, structure-based mapping revealed that the P-4G2 binding site overlaps with epitopes putatively targeted by neutralizing antibodies specific to HTNV and PUUV. These antibodies, derived following hantaviral infection, include mouse mAbs HC02 and 16E6 (*Arikawa et al., 1989*; *Wang et al., 1993*) that neutralize HTNV, and human mAb 1C9 (*Hörling and Lundkvist, 1997*; *Lundkvist et al., 1993a*), which neutralizes PUUV (*Figure 3*). Thus, while our identified epitope appears to be a common target of the neutralizing antibody response, future investigations into the molecular basis of immune recognition will undoubtedly be required to fully define the antigenic topography of Gn and Gc. Such investigations will also likely clarify whether the neutralizing epitope defined here is similarly targeted upon immunization, and whether the Ab-immune response is differentially focused against Gn and Gc between reservoir species (e.g. bank vole) and accidental (i.e. humans) hosts.

Lattice integrated and lattice-free Gc are expected to transiently exist on the hantavirus envelope: contacts between $(Gn-Gc)_4$ spikes are mediated by Gc homo-dimers, while lattice breaks display Gc that is not engaged in discernible homotypic interactions (*Bignon et al., 2019*; *Huiskonen et al., 2010*; *Li et al., 2016*). In our cryo-ET reconstruction, Fab P-4G2 was found to bind PUUV Gc only in the lattice-free state, and the frequency of lattice-free spikes was higher in the presence of Fab P-4G2. This observation demonstrates that P-4G2 reduces the frequency of Gc-mediated lattice formation. While the bivalent binding mode of full-length mAb P-4G2, with respect to the structurally characterized Fab P-4G2, may result in a different extent of lattice deformation, it is likely that both the mAb and Fab function by either active destabilization of Gc–Gc interfaces through competition with adjacent epitopes (*Figure 6—figure supplement 2*), or by sequestration of naturally occurring free spikes in a process that drives the equilibrium between homo-dimeric and lattice-free Gc toward the lattice-free state.

Characterization of the Fab P-4G2 epitope has important implications for rational therapeutic design efforts and indicates that lattice-free conformation of the Gc can be targeted by the antibody-mediated immune response that arises during infection. Indeed, although the development of recombinant immunogens that recreate the higher-order hantaviral $(Gn-Gc)_4$ lattice is desirable and would likely elicit a robust immune response, our data also indicate that a reverse vaccinology approach (*Burton, 2017*) focusing on recombinant and more minimal (monomeric) hantaviral Gn or Gc may also elicit anti-hantaviral nAbs. Indeed, such an approach has been successful in eliciting neutralizing antibodies against the glycoproteins of other bunyaviruses, including Rift Valley fever

virus (phlebovirus) (*Allen et al., 2018*) and Schmallenberg virus (orthobunyavirus) (*Hellert et al., 2019*).

At present, treatment and prevention options for hantaviral disease remain extremely limited due to the absence of approved therapeutics. However, strides are being made toward the development of efficacious treatments, as shown by the recent development of novel mAbs against ANDV (*Duehr et al., 2020*; *Garrido et al., 2018*), and a polyclonal antibody treatment against HTNV (*Hooper et al., 2014*), both of which demonstrate efficacy in animal models. *In toto*, our integrated structural and functional analysis provides a first structure-based blueprint for targeting the glycoprotein surface of this group of important emerging pathogens.

# Materials and methods

**Key resources table**

| Reagent type (species) or resource | Designation | Source or reference | Identifiers | Additional information |
|---|---|---|---|---|
| Gene (*Puumala orthohantavirus*) | Glycoprotein precursor (GPC); used in recombinant Gc production | GenBank | CAB43026.1 | Synthetic cDNA was produced by GeneArt, Life Technologies |
| Gene (*Puumala orthohantavirus*) | Glycoprotein precursor (GPC); used in PUUV VLP production | GenBank | CCH22848.1 | Synthetic cDNA was produced by GeneArt, Life Technologies |
| Gene (*Hantaan orthohantavirus*) | Glycoprotein precursor (GPC) | GenBank | AIL25321.1 | Synthetic cDNA was produced by GeneArt, Life Technologies |
| Gene (*Andes orthohantavirus*) | Glycoprotein precursor (GPC) | GenBank | AAO86638.1 | Synthetic cDNA was produced by GeneArt, Life Technologies |
| Strain, strain background (*Escherichia coli*) | Subcloning Efficiency DH5α Competent Cells | Thermo Fisher Scientific | Cat#: 18265017 | Competent cells |
| Cell line (*Homo-sapiens*) | Human embryonic kidney HEK 293T | ATCC | CRL-3216 | |
| Cell line (*Homo-sapiens*) | Human embryonic kidney HEK293F | Thermo Fisher Scientific | Cat#: R79007 | |
| Biological sample (*Myodes Glareolus, Mus musculus*) | Bank vole-mouse heterohybridoma producing mAb P-4G2 | *Lundkvist and Niklasson, 1992* | | |
| Biological sample (*Indiana vesiculovirus*) | VSV-ΔG RFP | *Reynard and Volchkov, 2015* | | |
| Recombinant DNA reagent | pHLsec plasmid | *Aricescu et al., 2006* | | |
| Recombinant DNA reagent | pHLsec-8H-SUMO-1D4 plasmid | *Chang et al., 2015* | | |
| Recombinant DNA reagent | pgk-φC31/pCB92 plasmid | *Chen et al., 2011* | | |
| Recombinant DNA reagent | pURD plasmid | *Zhao et al., 2014* | | |
| Recombinant DNA reagent | Tim1/pCAGGs plasmid | *Watt et al., 2014* | | |
| Recombinant DNA reagent | pCAGGS plasmid | *Niwa et al., 1991* | | |

*Continued on next page*

*Continued*

| Reagent type (species) or resource | Designation | Source or reference | Identifiers | Additional information |
|---|---|---|---|---|
| Recombinant DNA reagent | Mouse IgG1 plasmid | *von Boehmer et al., 2016* | | |
| Recombinant DNA reagent | Mouse IgK plasmid | *von Boehmer et al., 2016* | | |
| Recombinant DNA reagent | Fab P-4G2 light chain synthetic DNA fragment | This study | | Synthetic cDNA was produced by GeneArt, Life Technologies |
| Recombinant DNA reagent | Fab P-4G2 heavy chain synthetic DNA fragment | This study | | Synthetic cDNA was produced by GeneArt, Life Technologies |
| Sequence-based reagent | Mouse IgG sequencing primers | *von Boehmer et al., 2016* | | List of primer sequences is provided in the referenced study |
| Sequence-based reagent | PUUV Gc ectodomain cloning primer, forward | This study | | CGCACCGGTGAGA CACAGAACCTGA ACAGCGGC |
| Sequence-based reagent | PUUV Gc ectodomain cloning primer, reverse | This study | | GCGGTACCCTCGCC GGACTTGGTGAACC |
| Sequence-based reagent | Fab P-4G2 HC R100A mutagenesis primer, forward | This study | | GTATTACTGTACAA GAGATGCATTA GGCCCTTTTGA |
| Sequence-based reagent | Fab P-4G2 HC R100A mutagenesis primer, reverse | This study | | TCAAAAGGGCCTAA TGCATCTCTTG TACAGTAATAC |
| Commercial assay or kit | Phusion High-Fidelity PCR Master Mix with HF Buffer | New England Biolabs | Cat#: M0531S | |
| Commercial assay or kit | SuperScript III reverse transcriptase | Thermo Fisher Scientific | Cat#: 18080093 | |
| Commercial assay or kit | Quick Ligation kit | New England Biolabs | Cat#: M2200S | |
| Chemical compound, drug | PEI Max 40K | Polysciences, Inc | Cat#: 24765–1 | |
| Chemical compound, drug | PEI | Polysciences, Inc | Cat#: 23966–1 | |
| Chemical compound, drug | Lipofectamine 2000 Transfection reagent | Thermo Fisher Scientific | Cat#: 11668027 | |
| Chemical compound, drug | Kifunensine | Cayman Chemical | Cat#: 10009437 | |
| Software, algorithm | XIA2 | *Winter, 2010* | | |
| Software, algorithm | CCP4 | *Potterton et al., 2003* | | |
| Software, algorithm | SWISS-MODEL | *Waterhouse et al., 2018* | | |
| Software, algorithm | Coot | *Emsley and Cowtan, 2004* | | |

*Continued on next page*

*Continued*

| Reagent type (species) or resource | Designation | Source or reference | Identifiers | Additional information |
|---|---|---|---|---|
| Software, algorithm | REFMAC | *Murshudov et al., 1997* | | |
| Software, algorithm | PHENIX | *Adams et al., 2002* | | |
| Software, algorithm | Molprobity | *Davis et al., 2007* | | |
| Software, algorithm | IMGT/V-QUEST server | *Brochet et al., 2008* | | |
| Software, algorithm | GraphPad Prism | GraphPad Software, San Diego, CA, USA | | |
| Software, algorithm | PDBePISA server | *Krissinel and Henrick, 2007* | | |
| Software, algorithm | PyMOL | The PyMOL Molecular Graphics System, Schrödinger, LLC | | |
| Software, algorithm | UCSF Chimera | *Pettersen et al., 2004* | | |
| Software, algorithm | LigPlot+ software | *Laskowski and Swindells, 2011* | | |
| Software, algorithm | Motioncor2 | *Zheng et al., 2017* | | |
| Software, algorithm | CTFFIND4 | *Rohou and Grigorieff, 2015* | | |
| Software, algorithm | *tomo_preprocess* script | This study | | |
| Software, algorithm | IMOD | *Mastronarde and Held, 2017* | | |
| Software, algorithm | Dynamo | *Castaño-Díez et al., 2012* | | |
| Software, algorithm | PatchFinder script | This study | | |
| Other | Chromatography column, Superdex 200 10/300 Increase | Cytiva | Cat#: 28990944 | |
| Other | Chromatography column, HisTrap FF Crude 5 ml | Cytiva | Cat#: 17528601 | |
| Other | 1-MDa cut-off dialysis membrane | Spectrum Chemical | | |
| Other | Holey carbon grids, 2 µm hole diameter | Protochips | | |

## Sequencing mAb P-4G2 variable regions from hybridoma cell line

RNA from the P-4G2 hybridoma cell line was converted into cDNA (SuperScript III reverse transcriptase, Life Technologies) using random hexamers following the manufacturer's protocol. The bank vole antibody variable regions of heavy and kappa chains were PCR amplified using previously described mouse primers and PCR conditions (*von Boehmer et al., 2016*). PCR products were purified and cloned into mouse IgG expression plasmids (*von Boehmer et al., 2016*) using sequence and ligation independent cloning (SLIC) under ampicillin selection. Antibody variable regions were sequenced by Sanger sequencing, and CDRs were determined from the sequences with IMGT/V-QUEST (*Brochet et al., 2008*) server using *Mus musculus* as comparison group.

Antibody heavy and light plasmids were co-transfected at a 1:1 ratio into HEK 293F cells (Thermo Fisher Scientific) using PEI Max 40K (linear polyethylenimine hydrochloride, Polysciences, Inc), as previously described (*Zeltina et al., 2017*). Antibody supernatants were harvested 7 days following transfection and purified using protein G affinity chromatography following the manufacturer's protocol (GE healthcare). In order to validate the functional relevance of Arg100, the R100A mutation was introduced into the P-4G2 heavy chain plasmid using site directed mutagenesis (forward primer: GTATTACTGTACAAGAGATGCATTAGGCCCTTTTGA, reverse primer: TCAAAAGG GCCTAATGCA TCTCTTGTACAGTAATAC) and the mutation was verified using Sanger sequencing. Mab P-4G2 R100A mutant was expressed and purified as described above for WT mAb P-4G2.

## Generation of hantavirus-pseudotyped VSV-ΔG RFP

$8 \times 10^6$ HEK 293T (ATCC CRL-3216) cells were transfected with a plasmid encoding PUUV GPC (35 μg in pCAGGS vector) and Tim-1 (3 μg) using PEI (3:1 PEI:DNA, Polysciences, Inc). After 24 hr, cells were infected with a VSV-G pseudotyped VSV-ΔG RFP stock (40 μl). Infection was monitored using RFP expression and virus (first stock) was harvested when cells showed rounding and began to detach (typically 24–48 hr).

Two additional rounds of transfection and virus production were carried out. $15 \times 10^6$ HEK 293T cells were transfected with a plasmid encoding PUUV GPC, ANDV GPC, or HTNV GPC (100 μg, in pCAGGS vector) and Tim-1 (6 μg) using PEI (3:1 PEI: DNA, Polysciences, Inc). After 24 hr, cells were infected with 400 μl of 'first stock' virus. Infection was monitored using RFP expression and virus was harvested (second stock) when cells showed rounding and began to detach (48–72 hr). This process was repeated to generate 'third stock' virus which was used in subsequent neutralization assays.

## Hantavirus-pseudotyped VSV neutralization assay

Neutralizing activity was assessed using a single-round replication pseudovirus assay with HEK 293T target cells. Briefly, the antibody was serially diluted in a 96-well black, flat-bottom plate and preincubated with virus for 1 hr at 37°C. Cells at a concentration of 30,000/well were added to the virus–antibody mixture, and RFP quantified 72 hr following infection (Envision plate reader, Perkin Elmer). Dose–response curves were fitted using nonlinear regression (GraphPad Prism) to determine 50% inhibitory concentration ($IC_{50}$).

## Recombinant protein expression and purification

Codon-optimized synthetic cDNA (GeneArt, Life Technologies) coding for Fab P-4G2 light chain and Fab P-4G2 heavy chain were individually cloned into the pHLsec mammalian expression vector and co-transfected into HEK 293T cells (ATCC CRL-3216) for transient expression, as previously described (*Avanzato et al., 2019*). Fab P-4G2-containing supernatant was collected, clarified by centrifugation, and diafiltrated using the ÄKTA Flux tangential flow filtration system. Diafiltrated cell supernatant was purified by immobilized nickel affinity chromatography (5 ml fast flow crude column and ÄKTA fast protein liquid chromatography [FPLC] system; GE Healthcare) at room temperature, using 250 mM imidazole for elution. Finally, the sample was purified by size exclusion chromatography (SEC) using a Superdex 200 10/300 Increase column (GE Healthcare), in 10 mM Tris (pH 8.0)–150 mM NaCl buffer.

In order to produce PUUV Gc, a HEK 293T cell line stably expressing the protein was generated. PUUV Gc residues 659–1105 (GenBank accession no. CAB43026.1), modified to contain a 3C protease cleavable SUMO tag and hexahistidine tag at the N-terminus and a 1D4 tag in the C-terminus (*Chang et al., 2015*), were cloned into the pURD vector (*Zhao et al., 2014*). The stable cell line was generated following the method published by *Seiradake et al., 2015*; in short, HEK 293T cells were transfected with the pURD vector containing the Gc insert, along with an integrase expression vector (pgk-φC31/pCB92) (*Chen et al., 2011*), followed by consecutive rounds of cultivation under puromycin selection. For crystallization, PUUV Gc-producing cells were cultivated in the presence of kifunensine (*Chang et al., 2007*), and purification was performed via successive diafiltration, nickel affinity, and SEC, as described above for Fab P-4G2. Oligomannose glycans derived from expression in the presence of kifunensine were retained in the purified Gc sample.

Prior to crystallization, the N-terminal SUMO and hexahistidine tags were cleaved from the PUUV Gc (1:10 molar ratio of protein to 3C protease added to purified sample, followed by incubation at

21°C for 12 hr). PUUV Gc sample was then mixed with pure Fab P-4G2 in 1:1.2 molar ratio, and the complex was purified by SEC using a Superdex 200 10/300 Increase column (GE Healthcare), in 10 mM Tris (pH 8.0)–150 mM NaCl buffer.

## Crystallization and structure determination

A solution containing a complex of PUUV Gc ectodomain and Fab P-4G2 was crystallized by sitting drop vapor diffusion using 100 nl protein (1.3 mg/ml) with 100 nl precipitant (20% vol/vol PEG 6000, 0.1 M MES pH 6, and 1 M lithium chloride) and 100 nl additive (6% 2-methyl-2,4-pentanediol) at room temperature. Crystals formed over 2 weeks and were flash frozen by immersion into a cryo-protectant containing 25% (vol/vol) glycerol followed by rapid transfer into liquid nitrogen. X-ray diffraction data were recorded at beamline I24 ($\lambda$ = 0.9686 Å), Diamond Light Source, United Kingdom.

Data were indexed and integrated with XIA2/XDS (*Winter, 2010*) and scaled using CCP4/SCALE-PACK2MTZ. Processing statistics are presented in *Supplementary file 1*. The structure of Fab P-4G2−PUUV Gc was solved by molecular replacement in Phenix-MR (*Adams et al., 2002*) using PUUV Gc (PDB 5J81) and a homology model of the Fab P-4G2, generated using the SWISS-MODEL server (*Waterhouse et al., 2018*) with PDB 6BPB as the template structure, as search models. Structure refinement was performed by iterative refinement using REFMAC and Phenix (*Adams et al., 2002*). Coot (*Emsley and Cowtan, 2004*) was used for manual rebuilding and MolProbity (*Davis et al., 2007*) was used to validate the model.

## Cell lines

Authenticated cell lines were obtained from ATCC (HEK 293T cells) or, in the case of FreeStyle 293F cell line, from the manufacturer Thermo Fisher, and were tested for mycoplasma contamination and found negative.

## Molecular graphics and protein interface analysis

Molecular graphics images were generated using PyMOL (The PyMOL Molecular Graphics System, Version 1.7.0.3, Schrödinger, LLC) and UCSF Chimera (*Pettersen et al., 2004*). Interfacing and interacting residues were identified using the PDBePISA server (*Krissinel and Henrick, 2007*) run with default parameters, where the maximum allowable distance between the heavy atoms of a hydrogen bond donor and acceptor was limited to 3.89 Å, and the angle between acceptor-hydrogen-donor atoms was limited to between 90 and 270°. The maximal allowable distance between heavy atoms of a salt bridge was 4.0 Å. LigpPlot+ (*Laskowski and Swindells, 2011*) was used to visualize the Fab P-4G2−Gc interface (*Figure 2—figure supplement 2*).

## Preparation of PUUV VLPs

PUUV VLPs were produced by transient expression of the complete PUUV M-segment (GenBank CCH22848.1) cloned into the pHLsec vector (*Aricescu et al., 2006*) in HEK 293T cells. Six five-layer 875 cm² flasks (Falcon) were used to produce 750 ml of VLP-containing media that was clarified at 3000 × g for 20 min to remove cell debris and filtered through a 0.45 µm filter. The virus containing medium was concentrated down to approximately 30 ml using a pump-powered filter (100 kDa cut-off; Vivaflow, Sartorius) and then dialyzed into an excess of buffer (10 mM Tris pH 8.0–150 mM NaCl) through a 1-MDa cut-off dialysis membrane (Biotech CE Tubing, Spectrum Chemical) for a few days. The media was further concentrated to ~3 ml with a 100 kDa cut-off centrifugal concentrator filter (Amicon Ultra, Merck Millipore) and layered onto a 20–60% w/v sucrose density gradient in PBS buffer. The gradient was prepared using a Gradient Master (BioComp Instruments, Canada) in a SW32 Beckman tube, and the VLP banded by ultracentrifugation at 4°C for 4 hr at 25,000 rpm. The diffuse band (volume of ~3–4 ml) was collected manually, diluted to 20 ml with PBS and pelleted through a cushion of 10% sucrose in PBS to further clean and concentrate the sample (SW32 Beckman centrifuge tube at 25,000 rpm at 4°C for 2 hr). Finally, the pellet was resuspended in 60 µl of PBS and stored at 4°C.

## Cryo-EM grid preparation, data acquisition, and data processing

A 3 µl aliquot of VLP sample supplemented with 3 µl of 6 nm gold fiducial markers (Aurion) was applied on a holey carbon grid (2 µm hole diameter, C-flat, Protochips) that had been glow discharged in a plasma cleaner (Harrick) for 15 s. The grids were blotted for 3–4 s and plunged into an ethane/propane mixture using a vitrification apparatus (Vitrobot, Thermo Fisher Scientific). For Fab P-4G2-treated VLP EM sample preparation, a suspension of purified VLPs was incubated with 2.71 µM Fab P-4G2 for 1 hr at room temperature prior to grid preparation.

Data were collected using a Titan Krios transmission electron microscope (Thermo Fisher Scientific) operated at 300 kV and at liquid nitrogen temperature. Tomo4 software was used to acquire tomographic tilt data on a direct electron detector (K2 Summit; Gatan) mounted behind an energy filter (0–20 eV; QIF Quantum LS, Gatan). Tilt series were collected from –30 to 60° in 3° increments with a dose of 4.5 e$^-$/Å$^2$ per tilt and six frames in each movie (*Supplementary file 1*).

Movie frames were aligned and averaged using Motioncor2 to correct for beam induced motion (*Zheng et al., 2017*). Contrast transfer function (CTF) parameters were estimated using CTFFIND4 (*Rohou and Grigorieff, 2015*) and a dose-weighting filter was applied to the images according to their accumulated electron dose as described previously (*Grant and Grigorieff, 2015*). These preprocessing steps were carried out using a custom script named *tomo_preprocess* (*Stass, 2020*; available at https://github.com/OPIC-Oxford/TomoPreprocess). Tilt images were then aligned using the fiducial markers, corrected for the effects of CTF by phase flipping and used to reconstruct 3D tomograms in IMOD (*Mastronarde and Held, 2017*). The amplitudes of the subvolumes cropped from the tomograms were weighted to correct for the low-pass filtering function resulting from dose-weighting of the original images using a custom script (available at https://github.com/OPIC-Oxford/TomoPreprocess).

Subvolume averaging was performed in Dynamo following previously established procedures (*Huiskonen et al., 2014*; *Li et al., 2016*). Briefly, initial particle locations ('seeds') were created by modeling the VLP membrane surface using the Dynamo's *tomoview* program. Subvolumes extracted from these locations were iteratively refined against their rotational average, calculated around the membrane model surface normal at each position, to accurately align their locations onto the membrane. Subsequent refinements were carried out using a map of the TULV GP spike (EMDB-4867) as an initial template. The template was low-pass filtered to 50 Å frequency to avoid model bias. Overlapping particles were removed based on a distance filter (106 Å) and cross-correlation threshold after each iteration.

A custom script (*Stass, 2020*; available at https://github.com/OPIC-Oxford/PatchFinder) was used to divide the spikes into those that were part of a lattice and those that were not. The script locates the eight closest spikes for each particle and iterates through each with two checks. First, the position of the neighbor must be within a given distance (49 Å) of the ideal values determined from the positions of neighbors in the previously reported hantaviral reconstruction (EMD code; EMD-4867) and the orientation of the neighbor must match the parent particle within a set tolerance (25°). Spikes were defined as being part of a lattice if they had at least three interacting neighbors. This classification approach allowed reconstructing the structure of the lattice-bound spike for both VLP datasets (with and without Fab P-4G2) in addition to a lattice-free, P-4G2-bound spike in the case of the Fab P-4G2 sample (*Supplementary file 1*). In the latter case, the refinement was repeated using P-4G2-bound spike as a starting model and by excluding weakly correlating particles (below 0.15). This led to a reconstruction with a lattice-free spike fully decorated with Fab P-4G2 around its perimeter (*Supplementary file 1*). The final maps were filtered to the resolution determined by FSC (0.143 threshold) and rendered as isosurfaces. The surface threshold value was determined according to the molecular weight of the viral ectodomain proteins (and additional Fab) assuming an average protein density of 0.81 Da/Å$^3$.

## Quantification of the frequency of lattice formation on VLP surface in the presence and absence of Fab P-4G2

The positions of each spike and the number of interacting neighbors were analyzed in PUUV VLPs in the presence and absence of Fab P-4G2 from eight VLPs from each cryo-ET dataset. To facilitate the analysis, only VLPs displaying good glycoprotein coverage and roughly spherical morphology were included. Both datasets were processed using the workflow described above, and the PatchFinder

script was applied to locate those spikes that were part of a lattice. The particles on the top and bottom of the virus were then removed from further analysis by restricting the second Euler angle (tilt) to 45–135°. This was done in order to exclude those spikes that are facing the air–water interface, which can in itself result in a loss of visible lattice due to its denaturing effect. The number of neighbors for each spike was quantified and the 3D locations of the spikes were plotted in 2D using Mercator projections (*Figure 4—figure supplement 2*).

## Fitting of the crystal structure into the cryo-EM map

The crystal structure of Fab P-4G2−PUUV Gc was docked into the reconstruction of the Fab-decorated PUUV Gn–Gc spike using the fitmap function in Chimera (*Pettersen et al., 2004*). A map was simulated for the crystal structure to the resolution of 13.4 Å, to match the resolution of the spike reconstruction, and fitting was performed using an initial global search followed by a local fitting that accounts for the C4 symmetry of the spike. The top solution (presented in *Figure 5*) yielded a local map-to-map correlation score of 0.91 (*Supplementary file 1*). In order to complete our model of the hantaviral spike, we then added PUUV Gn to the fit based on the recently reported ANDV Gn−Gc complex crystal structure (PDB 6Y5F) that describes the native Gn−Gc assembly. In short, MatchMaker function in Chimera was used to superpose ANDV Gn−Gc crystal structure on the docked Fab P-4G2−PUUV Gc structure, and PUUV Gn (PDB 5FXU) was then similarly superposed on the ANDV Gn−Gc structure. Models of PUUV Gc (docked) and PUUV Gn (placed according to the known Gn−Gc assembly from PDB 6Y5F) were then combined in order to generate a Gn−Gc model that could be reliably fitted into a reconstruction of lattice-incorporated PUUV VLP surface, where no Fab P-4G2 density is present. Local fitting of this PUUV Gn−PUUV Gc assembly into a reconstruction of lattice-incorporated PUUV VLP surface, using the fitmap function in Chimera, yielded a correlation score of 0.89 (PUUV VLP without Fab; map for structure simulated to 13.9 Å resolution). C4 symmetry was then applied to generate $(Gn–Gc)_4$ spike assemblies, which were docked to the densities of the central spike and the neighboring spikes in Chimera. Finally, clashes at spike interfaces were resolved by sequential fitting of the interfacing components. The resulting model of the Gn–Gc assemblies that comprise the hantaviral lattice is presented in *Figure 6*.

## Accession codes

Atomic coordinates and structure factors of the PUUV Gc−Fab P-4G2 complex have been deposited in the PDB (accession code 6Z06). Cryo-EM reconstructions of the PUUV VLP surface alone and in the presence of Fab P-4G2 from areas of continuous and discontinuous lattice have been deposited in the EMDB at the EBI under accession codes EMD-11966, EMD-11965, and EMD-11964, respectively. Coordinates of protein structures fitted into these cryo-EM reconstructions have been deposited in the PDB database (accession code 7B09 for the Fab P-4G2-PUUV Gc and PUUV Gn fitted into EMD-11964, and accession code 7B0A for PUUV Gc and PUUV Gn fitted into EMD-11966, respectively).

## Acknowledgements

We thank Diamond Light Source for beamtime (proposal MX19946) and the staff of beamline I24 at Diamond Light Source for assistance with data collection. We thank the MRC (MR/L009528/1 and MR/S007555/1 to TAB; MR/N002091/1 to KJD and TAB; MR/K024426/1 to KJD), Academy of Finland (309605 to IR), and European Research Council under the European Union's Horizon 2020 research and innovation programme (649053 to JTH). The Wellcome Centre for Human Genetics is supported by Wellcome Centre grant 203141/Z/16Z. We thank Dimple Karia, Bilal Qureshi, and Adam Costin for electron microscopy support. The Oxford Particle Imaging Centre was founded by a Wellcome Trust JIF award (060208/Z/00/Z) and is supported by equipment grants from WT (093305/Z/10/Z) and MRC. We acknowledge the use of Central Oxford Structural Molecular Imaging Centre (COSMIC) facilities and CSC – IT Center for Science, Finland, for computational resources. Computation also used the Oxford Biomedical Research Computing (BMRC) facility, a joint development between the Wellcome Centre for Human Genetics and the Big Data Institute supported by Health Data Research UK and the NIHR Oxford Biomedical Research Centre. The views expressed are those of the author(s) and not necessarily those of the NHS, the NIHR, or the Department of Health.

## Additional information

### Funding

| Funder | Grant reference number | Author |
|---|---|---|
| Medical Research Council | MR/L009528/1 | Thomas A Bowden |
| Medical Research Council | MR/S007555/1 | Thomas A Bowden |
| Medical Research Council | MR/N002091/1 | Thomas A Bowden<br>Katie J Doores |
| Medical Research Council | MR/K024426/1 | Katie J Doores |
| Academy of Finland | 309605 | Ilona Rissanen |
| H2020 European Research Council | 649053 | Juha T Huiskonen |
| Wellcome Trust | 203141/Z/16Z | Robert Stass<br>Ruben JG Hulswit<br>Guido C Paesen<br>Juha T Huiskonen<br>Thomas A Bowden |
| Wellcome Trust | 060208/Z/00/Z | Robert Stass<br>Ruben JG Hulswit<br>Guido C Paesen<br>Juha T Huiskonen<br>Thomas A Bowden |
| Wellcome Trust | 093305/Z/10/Z | Robert Stass<br>Ruben JG Hulswit<br>Guido C Paesen<br>Juha T Huiskonen<br>Thomas A Bowden |

The funders had no role in study design, data collection and interpretation, or the decision to submit the work for publication.

### Author contributions

Ilona Rissanen, Conceptualization, Formal analysis, Funding acquisition, Investigation, Visualization, Writing - original draft, Writing - review and editing; Robert Stass, Investigation, Writing - original draft, Writing - review and editing; Stefanie A Krumm, Jeffrey Seow, Ruben JG Hulswit, Guido C Paesen, Investigation, Writing - review and editing; Jussi Hepojoki, Resources, Investigation, Writing - review and editing; Olli Vapalahti, Writing - review and editing; Åke Lundkvist, Resources, Writing - review and editing; Olivier Reynard, Viktor Volchkov, Resources, Methodology, Writing - review and editing; Katie J Doores, Thomas A Bowden, Conceptualization, Supervision, Funding acquisition, Investigation, Writing - original draft, Writing - review and editing; Juha T Huiskonen, Conceptualization, Supervision, Funding acquisition, Investigation, Writing - review and editing

### Author ORCIDs

Juha T Huiskonen https://orcid.org/0000-0002-0348-7323
Thomas A Bowden https://orcid.org/0000-0002-8066-8785

### Decision letter and Author response

Decision letter https://doi.org/10.7554/eLife.58242.sa1
Author response https://doi.org/10.7554/eLife.58242.sa2

## Additional files

### Supplementary files

• Supplementary file 1. Table S1: Crystallographic data collection and refinement statistics for Fab P-4G2—Puumala virus Gc. Table S2. Cryo-EM tomography data collection, sub-tomogram reconstruction, and fitting statistics.

• Transparent reporting form

### Data availability

Atomic coordinates and structure factors of the PUUV Gc-Fab P-4G2 complex crystal structure have been deposited in the PDB under accession code 6Z06. Cryo-EM reconstructions of the PUUV VLP surface alone and in the presence of Fab P-4G2 from areas of continuous and discontinuous lattice have been deposited in the EMDB at the EBI under accession codes EMD-11966, EMD-11965 and EMD-11964, respectively. Coordinates of protein structures fitted into these cryo-EM reconstructions have been deposited in the PDB database (accession code 7B09 for the Fab P-4G2-PUUV Gc and PUUV Gn fitted into EMD-11964, and accession code 7B0A for PUUV Gc and PUUV Gn fitted into EMD-11966, respectively). The following authors authored the data deposited to PDB and EMDB, as described above.

The following datasets were generated:

| Author(s) | Year | Dataset title | Dataset URL | Database and Identifier |
|---|---|---|---|---|
| Rissanen IR, Stass R, Krumm SA, Seow J, Hulswit RJG, Paesen GC, Hepojoki J, Vapalahti O, Lundkvist A, Reynard O, Volchkov V, Doores KJ, Huiskonen JT, Bowden TA | 2020 | Crystal structure of Puumala virus Gc in complex with Fab 4G2 | https://www.rcsb.org/structure/6Z06 | RCSB Protein Data Bank, 6Z06 |
| Rissanen IR, Stass R, Huiskonen JT, Bowden TA | 2020 | Puumala virus glycoprotein (Gc) in complex with fab fragment P-4G2 | https://www.rcsb.org/structure/7B09 | RCSB Protein Data Bank, 7B09 |
| Rissanen IR, Stass R, Huiskonen JT, Bowden TA | 2020 | Puumala virus-like particle glycoprotein spike and lattice contacts model | https://www.rcsb.org/structure/7B0A | RCSB Protein Data Bank, 7B0A |
| Rissanen IR, Stass R, Huiskonen JT, Bowden TA | 2020 | Puumala virus-like particle glycoprotein lattice | http://www.ebi.ac.uk/pdbe/entry/emdb/EMD-11966 | Electron Microscopy Data Bank, EMD-11966 |
| Rissanen IR, Stass R, Huiskonen JT, Bowden TA | 2020 | A reconstruction of Puumala virus-like particle glycoprotein spike lattice in the presence of fab 4G2 which is absent from the reconstruction | http://www.ebi.ac.uk/pdbe/entry/emdb/EMD-11965 | Electron Microscopy Data Bank, EMD-11965 |
| Rissanen IR, Stass R, Huiskonen JT, Bowden TA | 2020 | Puumala virus-like particle glycoprotein spike in complex with fab fragment P-4G2 | http://www.ebi.ac.uk/pdbe/entry/emdb/EMD-11964 | Electron Microscopy Data Bank, EMD-11964 |

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
