## [Decision Letter]

Thank you for submitting your article "A fusion glycoprotein-targeting neutralizing antibody reveals the dynamic landscape of the hantavirus surface" for consideration by *eLife*. Your article has been reviewed by three peer reviewers, one of whom is a member of our Board of Reviewing Editors, and the evaluation has been overseen by José Faraldo-Gómez as the Senior Editor. The following individual involved in review of your submission has agreed to reveal their identity: Yorgo Modis (Reviewer #2).

The reviewers have discussed the reviews with one another and the Reviewing Editor has drafted this decision to help you prepare a revised submission.

Our expectation is that the authors will eventually carry out additional experiments and report on how they affect the relevant conclusions either in a preprint on bioRxiv or medRxiv, or if appropriate, as a Research Advance in *eLife*, either of which would be linked to the original paper.

Summary:

In this study, Rissanen et al. cloned the variable heavy and kappa (VH and VK) chains of a previously-described neutralizing monoclonal antibody against Puumala virus (PUUV), antibody 4G2 from the bank vole. The VH and VK chains were fused with mouse constant regions and recombinant chimeric Fab fragments were expressed and purified. The authors show that the 4G2 Fab fragment potently neutralizes PUUV in a hantavirus-pseudotyped VSV-ΔG RFP neutralization assay. Lower levels of neutralization against a virus pseudotyped with Andes virus (ANDV) Gc were also observed. A crystal structure of the Fab bound to PUUV Gc was determined. PUUUV Gc was found to be in a pre-fusion-like conformation. The 4G2 epitope – at the interface between domains I and II – included residues that are buried in the post-fusion trimeric Gc. The authors conclude that mAb 4G2 is specific to the pre-fusion state and that binding of the antibody is likely to sterically preclude the formation of a fusogenic configuration. The authors went on to generate PUUV virus-like particles (VLPs) and determine a cryoEM tomography image reconstruction with subtomogram averaging of Gc spikes on purified VLPs. Free hetero-octameric Gn-Gc spikes were found to have 4G2 Fabs bound (one on each Gc protomer), whereas spikes that were part of a spike lattice lacked density for the antibody fragment. Moreover, the frequency of free spikes was higher in the presence of Fab 4G2. The crystal structure of the Gc-Fab complex could be fitted into the cryoEM reconstruction, revealing the position of Gc within the cryoEM density for the spike.

This study identifies the epitope of neutralizing antibody 4G2, establishes a method for production of recombinant 4G2 Fab fragments, and identifies the mechanism of neutralization by 4G2 as inhibition of the fusogenic conformational change. Combination of the Fab-Gc crystal structure and the cryoEM reconstruction of PUUV VLPs allows the atomic model of the Gc-Fab complex to be fitted into the cryoEM density, revealing the position of Gc within the Gn-Gc spike density.

If accurate, this will contribute substantially to the molecular understanding of the structural organization of PUUV Gc protein. Overall, the study is interesting and generally sound, and with some revisions, would be appropriate to publish in *eLife*.

Essential revisions:

1) Title. There are no measurements of dynamics in this paper. While the authors discuss the possibility of dynamics, there is no evidence that the Fab captures dissociating surface protein or that Fab binding causes the dissociation, nor is there direct evidence of dynamics on an intact PUUV virus. Therefore changing the title from "…reveals the dynamic landscape" to something along the lines of "…reveals the conformational landscape" may be more appropriate.

2) It is not clear what structures have already been solved with regards to Fab/Gc complexes for hantaviruses and what novel value this particular crystal structure has in relation to published structures. Is this the first example of an antibody bound to a PUUV surface protein? Is this the first example of a host antibody bound to any hantavirus Gc? If so, the authors should explicitly state these things in their paper.

3) In the Introduction, it is unclear when the authors are talking about humans vs. other animals. Generalizations of human and vole immune responses as being the same may not be correct. The authors do a good job of not generalizing in their Discussion, but they don't make this distinction in the Introduction. This paragraph in the Introduction would benefit from being rewritten to be clearer.

4) It is hard to understand the interaction between the Fab and Gc protein in the crystal structure in Figure 2. In the zoomed-in panels in Figure 2B, there are unlabeled side chains shown and a surface rendering for the Gc protein, so it's impossible to see what the pointed-out side chain residues might be interacting with or which residues are which. A clearer figure depicting interaction and with labeled side chains is needed.

The authors include Figure 2—figure supplement 1 showing a 2Fo-Fc electron density map of part of the interaction region. A 2Fo-Fc simulated annealing omit map of the density for the interaction interface should be shown to eliminate the possibility of phase bias. The electron density looks as though it might benefit from (additional) map sharpening, as is often the case for lower-resolution crystal structures. The authors should try different B-factor sharpening corrections and show the density with the most informative correction, stating the correction factor applied.

In addition, what parameters do the authors use to define interactions for side chains (i.e., H-bonds, salt bridges, etc.)? The authors should give actual values of cut-offs for distances and angles being used in these definitions and not just say they used PDBePISA.

5) The authors conclude from their overlay analysis shown in Figure 3—figure supplement 1 that the crystal structure of PUUV Gc they solved is in the pre-fusion conformation. The orientation of domain III in the PUUV prefusion Gc structure reported here is very different from previously reported HTNV prefusion Gc structure (Figure 3—figure supplement 1). This means that domain III is likely to undergo a much bigger structural translocation to transition into the postfusion form than what was previously imagined. The flexibility of domain III should be added to the Discussion.

In addition, the hantavirus Gn-Gc proteins are expected to be tetramers on the surface of cells, but the authors state that their crystal structure of Gc is in monomeric form and that none of the structures they overlay were solved in a tetrameric form, reflecting what is proposed later in this study. The loop connecting Domain III to the rest of the Gc protein looks like it would be flexible. All of the structures they overlay with theirs in Figure 3—figure supplement 1 have crystal contacts between Domain III and neighboring molecules. Without a higher order interaction or the presence of Gn (where applicable), it could also be possible that Domain III has a larger range of motion in all of these crystal structures than in vivo due to the lack of other subunits and may be positioned in a non-biologically-relevant orientation as a consequence of crystal packing.

From the authors' crystal structure, is there evidence of crystal contacts with Domain III? If so, how do the authors rule out that their Gc-Fab crystal structure's Domain III position is not an artifact of crystal packing and is instead the biologically relevant pre-fusion orientation? Again, it's hard to conclude from their docked structure in Figure 5A if the Domain III actually fits or doesn't fit into their density because they only show a side view.

6) I find it hard to understand how neighbors were defined for surface proteins in the lattice analysis (Figure 4 and Figure 4—figure supplement 2). Can the authors clarify for an average reader what was done and what parameters were used?

7) The fitting of Gc atomic model in the cryoEM density (subsection “Mab 4G2 is specific to a monomeric state of virion-displayed PUUV Gc” and Figure 5) is an important result that is underplayed in this manuscript. Specifically, the fit presented in Figure 5 could easily be used to model the inter-spike Gc-Gc dimer interface, i.e. the lattice interface. This has been suggested to be an important interface (see for example Bignon et al., 2019), and the authors also propose that 4G2 binding breaks up the interface so it would be important to show a molecular model combining available cryoEM STA data for in-lattice spike and the fit in Figure 5, excluding the Fab fragments.

8) Top views of the sub-volume average of the single Gn-Gc surface protein are not shown (see Figure 4 and Figure 5). Why? A top-down orientation view would improve the reader's understanding of the organization of the structure and would allow for visual assessment of the quality of the fitting, which is not possible with only a side view.

9) For the lattice structure, how many neighbors did individual spikes have in the data used to make the lattice reconstruction in Figure 4B? Also, please comment on why the error bars are so large for the +4G2 spikes with 0, 1, or 2 neighbors in your chart in Figure 4B. This chart shows that there were examples in the data of spikes with less than 8 neighbors. Are the authors able to make reconstructions of spikes with less than 8 neighbors using subsets of the data? What would those look like? How many surface proteins in their VLPs are bound with Fab? How did the authors pick the concentrations for making their Fab/VLP samples?

10) Overall, the paper is missing some numbers that are necessary sanity checks for structures as well as the proposed mechanisms.

a) How big are PUUV virions? How big are the VLPs?

b) What is their total surface area of the VLPs? How many surface particles are on the VLPs?

c) How much space is left on the surface of the virion for particles to dissociate? In other words, does the 3D space on the VLP surface make sense for the spreading out of the proteins with the Fabs bound? Figure 4—figure supplement 2 looks like the surface is very saturated on the VLP alone examples (left), especially in the tomographic slice.

d) Why do the non-Fab VLPs look like they have more surface protein (Figure 4—figure supplement 2, left panels) than the Fab-bound VLPs (Figure 4—figure supplement 2, right panels)?

e) How big is the reconstructed patch of lattice vs. the total surface area available for the virus? What portion of the viral surface do they predict to be occupied by the surface proteins in a lattice organization?

11) It will be unclear to most readers what the "metastable homodimerization interface" is and how it fits into the virus lifecycle. This needs further context in the Introduction, or earlier in the Results (similar to the statement in the fourth paragraph of the Discussion). Related to this, the authors to clarify in the text that single spikes consist of (Gn-Gc)4 hetero-octamers. Related to point above, "Gc only in the monomeric state", and references to the monomeric state elsewhere in the text, are confusing because Gc is part of a heterooctameric Gn-Gc spike. Clarification is required to distinguish between Gc-dimerizing lattice interactions and intra-spike interactions.

On a similar note, in the Discussion, the authors say "The identification of the Gc oligomerization-specific epitope…" This wording is potentially misleading. Calling it “the Gc…” makes it sound like this is the only possible epitope for an antibody that would disrupt oligomerization. This may not be true. In addition, the phrase "oligomerization-specific epitope" makes it sound like this is the endogenous interaction site for the oligomerization between trimers to form a lattice, which the authors don't have any proof for because they don't dock structures into the oligomer lattice map that would potentially show the interfaces between trimers, nor do they make any mutations to disrupt the interaction to show that the lattice goes away on a VLP without the antibody present. Rephrasing to something like “The identification of an epitope that disrupts trimer oligomerization…” might be more appropriate.

12) There is a discrepancy in the phrasing between the Abstract and the main text as follows. In the Abstract, authors say, "analysis demonstrates that 4G2 binds to a multi-domain site on the PUUV Gc and precludes fusogenic rearrangements…" In the main text, the authors say "may sterically preclude the formation of a fusogenic configuration…" I find that their data does not concretely determine that the fab binding does preclude rearrangements, so the phrasing may preclude from their main text is most appropriate. The Abstract should be rewritten to match the phrasing in the main text.

---

## [Author Response]

Essential revisions:1) Title. There are no measurements of dynamics in this paper. While the authors discuss the possibility of dynamics, there is no evidence that the Fab captures dissociating surface protein or that Fab binding causes the dissociation, nor is there direct evidence of dynamics on an intact PUUV virus. Therefore changing the title from "…reveals the dynamic landscape" to something along the lines of "…reveals the conformational landscape" may be more appropriate.

We thank the reviewers for this helpful comment and agree that our manuscript does not provide direct evidence for dynamics on the hantavirus surface. In line with this comment, we have updated the title of our manuscript to:“Molecular rationale for antibody-mediated targeting of the hantavirus fusion glycoprotein”. Furthermore, we have removed reference to our manuscript defining the dynamic landscape of the hantavirus surface in the Abstract and Introduction section of our manuscript. We hope that the reviewers agree that this modified title and modification to the Introduction better reflects the main findings of our investigation.

2) It is not clear what structures have already been solved with regards to Fab/Gc complexes for hantaviruses and what novel value this particular crystal structure has in relation to published structures. Is this the first example of an antibody bound to a PUUV surface protein? Is this the first example of a host antibody bound to any hantavirus Gc? If so, the authors should explicitly state these things in their paper.

We thank the reviewer for the opportunity to clarify this point. While a non-neutralizing antibody has been used to facilitate crystallization of Hantaan virus (HTNV) Gc glycoprotein (Guardado-Calvo et al., 2016), our study is the first to report the structure of a hantavirus-neutralizing antibody. As recommended, we have revised our manuscript text to better highlight the novelty of the structure and results:

“Although elicitation of a productive nAb response constitutes an essential component of the host immune response to hantavirus infection (Arikawa et al., 1989; Heiskanen et al., 1999; Levanov et al., 2019; Lundkvist and Niklasson, 1992), there is a paucity of information with regards to the molecular basis of immune recognition.”

And

“This work provides first molecular-level insights into antibody-mediated neutralization of the hantavirus Gc and provides a rational basis for understanding immune-mediated targeting of the antigenic hantaviral surface.”

And

“In toto, our integrated structural and functional analysis provides a first structure-based blueprint for targeting the glycoprotein surface of this group of important emerging pathogens.”

3) In the Introduction, it is unclear when the authors are talking about humans vs. other animals. Generalizations of human and vole immune responses as being the same may not be correct. The authors do a good job of not generalizing in their Discussion, but they don't make this distinction in the Introduction. This paragraph in the Introduction would benefit from being rewritten to be clearer.

We thank the reviewer for the opportunity to clarify this point. While a non-neutralizing antibody has been used to facilitate crystallization of Hantaan virus (HTNV) Gc glycoprotein (Guardado-Calvo et al., 2016), our study is the first to report the structure of a hantavirus-neutralizing antibody. As recommended, we have revised our manuscript text to better highlight the novelty of the structure and results:

“Although elicitation of a productive nAb response constitutes an essential component of the host immune response to hantavirus infection (Arikawa et al., 1989; Heiskanen et al., 1999; Levanov et al., 2019; Lundkvist and Niklasson, 1992), there is a paucity of information with regards to the molecular basis of immune recognition.”

And

“This work provides first molecular-level insights into antibody-mediated neutralization of the hantavirus Gc and provides a rational basis for understanding immune-mediated targeting of the antigenic hantaviral surface.”

And

“In toto, our integrated structural and functional analysis provides a first structure-based blueprint for targeting the glycoprotein surface of this group of important emerging pathogens.”

4) It is hard to understand the interaction between the Fab and Gc protein in the crystal structure in Figure 2. In the zoomed-in panels in Figure 2B, there are unlabeled side chains shown and a surface rendering for the Gc protein, so it's impossible to see what the pointed-out side chain residues might be interacting with or which residues are which. A clearer figure depicting interaction and with labeled side chains is needed.

We are grateful for the opportunity to improve our depiction of the Fab 4G2−Gc interface. As requested, we have updated Figure 2B, to provide a more detailed view of the interaction at Arg100. Additionally, please note that we have also added our pseudovirus neutralization data demonstrating that Arg100 is functionally a key site on the antibody-antigen interface (panel C).

Furthermore, we have added an additional figure supplement, Figure 2—figure supplement 2, that provides greater detail for the interacting residues across the entire antibody−antigen interface. Furthermore, regarding supplementary materials, we would like to note that in agreement with *eLife* editorial policy, all supplementary figures have been re-titled figure supplements and linked to their relevant main figure item.

These new Figure 2 elements are referred to in revised Results section:

“Arg100 (CDRH3) is the most centrally located paratope residue, and is predicted to form multiple hydrogen bonds and a putative salt bridge with Gc Glu725 and across the antibody−antigen interface (Figure 2B and Figure 2—figure supplement 2; interacting residues were identified using the PDBePISA server (Krissinel and Henrick, 2007)). Indeed, site-directed mutagenesis of Arg100 to Ala considerably reduced the potency of P-4G2 to neutralize PUUV (Figure 2C).”

The authors include Figure 2—figure supplement 1 showing a 2Fo-Fc electron density map of part of the interaction region. A 2Fo-Fc simulated annealing omit map of the density for the interaction interface should be shown to eliminate the possibility of phase bias. The electron density looks as though it might benefit from (additional) map sharpening, as is often the case for lower-resolution crystal structures. The authors should try different B-factor sharpening corrections and show the density with the most informative correction, stating the correction factor applied.

We thank the reviewers for this helpful comment. As requested, we have calculated a simulated annealing composite omit map, which is now represented in revised Figure 2—figure supplement 1. We also note that we attempted map sharpening, however this did not produce visible improvement in the map.

In addition, what parameters do the authors use to define interactions for side chains (i.e., H-bonds, salt bridges, etc.)? The authors should give actual values of cut-offs for distances and angles being used in these definitions and not just say they used PDBePISA.

As requested, we have provided details for the prediction of hydrogen bonds and salt bridges by PDBePISA (as defined: https://www.ebi.ac.uk/msd-srv/prot_int/pi_tips.html), which were calculated using default parameters. The maximum allowable distance between the heavy atoms of a hydrogen bond donor and acceptor was 3.89 Å and the angle between acceptor-hydrogen-donor atoms was limited to between 90 and 270 degrees. Similarly, the distance between heavy atoms of a salt bridge for a salt bridge was 4.0 Å. These details have been added to the Materials and methods section, as follows:

“Molecular graphics and protein interface analysis. Molecular graphics images were generated using PyMOL (The PyMOL Molecular Graphics System, Version 1.7.0.3, Schrödinger, LLC) and UCSF Chimera (Pettersen et al., 2004). […] The maximal allowable distance between heavy atoms of a salt bridge was 4.0 Å. Ligplot+ (Laskowski and Swindells, 2011) was used to visualize the Fab P-4G2−Gc interface (Figure 2—figure supplement 2).”

5) The authors conclude from their overlay analysis shown in Figure 3—figure supplement 1 that the crystal structure of PUUV Gc they solved is in the pre-fusion conformation. The orientation of domain III in the PUUV prefusion Gc structure reported here is very different from previously reported HTNV prefusion Gc structure (Figure 3—figure supplement 1). This means that domain III is likely to undergo a much bigger structural translocation to transition into the postfusion form than what was previously imagined. The flexibility of domain III should be added to the Discussion.

The reviewers are correct that the translocation of domain III between pre and post-conformational states, as visualized in revised Figure 3 and Figure 3—figure supplement 1), is likely to be larger than originally appreciated in earlier structural studies of the hantavirus Gc (Guardado-Calvo et al., 2016; Willensky et al., 2016). We have revised Figure 3 to highlight the mobility of domain III in our structural analysis. Additionally, to further support that our observed conformation is in the prefusion state, we draw upon a recently reported Andes virus Gn-Gc crystal structure in the prefusion state (Serris et al., 2020), which presents a conformation of domain III that is highly similar to that observed in our own crystal structure. To reflect these observations, we have updated Figure 3—figure supplement 1 and added the following text to reflect this point in the Discussion section of the text:

“Our structural data provides several molecular-level insights into hantavirus Gc functionality and antibody-mediated neutralization. […] Similar structural intermediates have been identified in previous structural studies of related phlebovirus Gc glycoproteins, alone (Dessau and Modis, 2013) and in the context of the virion (Halldorsson et al., 2018), which is consistent with domain III flexibility being an intrinsic feature required for Gc glycoprotein functionality.”

In addition, the hantavirus Gn-Gc proteins are expected to be tetramers on the surface of cells, but the authors state that their crystal structure of Gc is in monomeric form and that none of the structures they overlay were solved in a tetrameric form, reflecting what is proposed later in this study. The loop connecting Domain III to the rest of the Gc protein looks like it would be flexible. All of the structures they overlay with theirs in Figure 3—figure supplement 1 have crystal contacts between Domain III and neighboring molecules. Without a higher order interaction or the presence of Gn (where applicable), it could also be possible that Domain III has a larger range of motion in all of these crystal structures than in vivo due to the lack of other subunits and may be positioned in a non-biologically-relevant orientation as a consequence of crystal packing.From the authors' crystal structure, is there evidence of crystal contacts with Domain III? If so, how do the authors rule out that their Gc-Fab crystal structure's Domain III position is not an artifact of crystal packing and is instead the biologically relevant pre-fusion orientation? Again, it's hard to conclude from their docked structure in Figure 5A if the Domain III actually fits or doesn't fit into their density because they only show a side view.

The reviewers brings up an excellent point that domain III flexibility appears to be an inherent feature of recombinantly produced class-II fusion glycoproteins, where the crystallographically observed conformations of domain III can likely be attributed to differential packing environments. Indeed, while our observed PUUV Gc domain III is also stabilized by crystal contacts, our confidence that the observed conformation closely corresponds to a pre-fusion state arises from the goodness of fit into the cryo-ET reconstruction of PUUV VLPs as well as the close structural correspondence between our PUUV Gc and the Gc of ANDV, in the Gn-Gc prefusion state (Serris et al., 2020). Additionally, we apologize for not better representing the fit of domain III in our original submission and have added panels to revised Figure 5C, which present close-up views of our fitting of this region into the reconstruction. We hope the reviewer’s agree that our observed conformation of domain III fits well into the membrane-proximal density of the VLP surface, and that the overall agreement of the crystallographic and cryoET data provides evidence that our PUUV Gc closely resembles a pre-fusion conformation.

To reflect the goodness of fit of PUUV domain III into the reconstruction, we have also added the following text to the Results section of the manuscript:

“PUUV Gc_Gc−P-4G2_ fits well within the cryo-ET reconstruction, where the crystallographically observed conformation of domain III accurately matches a feature of the glycoprotein spike (Figure 5C), further supporting that PUUV Gc has crystallized in the pre-fusion conformation presented on the mature virion.”

6) I find it hard to understand how neighbors were defined for surface proteins in the lattice analysis (Figure 4 and Figure 4—figure supplement 2). Can the authors clarify for an average reader what was done and what parameters were used?

We thank the reviewer for the opportunity to clarify this point. The refinement of pleomorphic virus glycoprotein lattices begins with a template matching procedure, which inevitably results in many false-positive “bad” particles that contribute noise within the dataset. The rationale for our so-called “PatchFinder” approach is to identify the “good” particles by checking that the position and orientation of the particles that surround a “parent” particle makes sense based on prior knowledge of the lattice. The script first identifies the 8 nearest particles for each parent particle, and then iterates through each neighbor with two checks. Firstly, PatchFinder checks that the position of a neighbor is within a list of possible positions in space. These positions are defined according to the observed positions of neighbors in a previously reported hantavirus reconstruction (EMD code; EMD-4867). A generous tolerance is allowed (49-Å) to account for both the flexibility of the lattice and for alignment inaccuracies. Secondly, Patchfinder checks whether the orientation of the neighboring particle, with respect to the parent particle, is within a defined angular tolerance (25-degrees). If at least three out of eight neighboring particles satisfy these conditions, the parent particle is accepted. Three is chosen as it would keep a parent particle that is located, for example, on the corner of a square patch of lattice and is ultimately a trade-off between eliminating enough bad particles, while keeping as many good particles as possible.

In our revision, we have clarified this decision-making process in the Materials and methods section of our revision as follows:

“The script locates the eight closest spikes for each particle, and iterates through each with two checks. Firstly, the position of the neighbour must be within a given distance (49-Å) of the ideal values determined from the positions of neighbours in the previously reported hantaviral reconstruction (EMD code; EMD-4867) and the orientation of the neighbour must match the parent particle within a set tolerance (25-degree).”

7) The fitting of Gc atomic model in the cryoEM density (subsection “Mab 4G2 is specific to a monomeric state of virion-displayed PUUV Gc” and Figure 5) is an important result that is underplayed in this manuscript. Specifically, the fit presented in Figure 5 could easily be used to model the inter-spike Gc-Gc dimer interface, i.e. the lattice interface. This has been suggested to be an important interface (see for example Bignon et al., 2019), and the authors also propose that 4G2 binding breaks up the interface so it would be important to show a molecular model combining available cryoEM STA data for in-lattice spike and the fit in Figure 5, excluding the Fab fragments.

We thank the reviewers for this opportunity to expand our structural analysis of the hantaviral surface. As requested, we have modelled the composition of the hantaviral glycoprotein lattice using our pre-fusion PUUV Gc structure (excluding Fab P-4G2) fitted into our 13.9- Å cryo-ET reconstruction of PUUV VLP surface. These results are presented in new Figure 6 and its figure supplements (Figure 6—figure supplements 1 and 2), and by the following addition to Results section of the manuscript:

“Drawing upon the fitting generated above, we used a 13.9-Å cryo-ET reconstruction of the lattice-incorporated PUUV VLP surface to create a model of the higher-order (Gn**−**Gc)_4_ spike lattice. […] Furthermore, modeling of Fab P-4G2 binding onto the Gc homodimer demonstrates that neighboring epitopes are not mutually accessible for Fab binding in their lattice-integrated form, providing clues to how Fab P-4G2 recognition may be accompanied by dissociation of the Gc homodimer assembly (Figure 6—figure supplement 2).”

Furthermore, we have updated the Materials and methods section to include the details of this extended fitting approach that included the placement of the Gn as follows:

“In order to complete our model of the hantaviral spike, we then added PUUV Gn to the fit based on the recently reported ANDV Gn−Gc complex crystal structure (PDB 6Y5F) that describes the native Gn−Gc assembly. […] The resulting model of the Gn-Gc assemblies that comprise the hantaviral lattice is presented in Figure 6.”

8) Top views of the sub-volume average of the single Gn-Gc surface protein are not shown (see Figure 4 and Figure 5). Why? A top-down orientation view would improve the reader's understanding of the organization of the structure and would allow for visual assessment of the quality of the fitting, which is not possible with only a side view.

We thank the reviewers for this helpful comment. We have now included a top view in revised Figure 5, which greatly improves visualization of the Gn-Gc glycoprotein organization with respect to the epitope. Furthermore, in response to this comment and comment 7, we have generated a new Figure 6, which presents both side-and top views of a PUUV VLP reconstruction displaying the continuous lattice formed from Gn_4_−Gc_4_ spikes. Furthermore, we have included the previously published crystal structure of PUUV Gn (PDB id: 5FXU) into our fitting, and present the Gn structure as part of the spike assembly in both revised Figure 5 and new Figure 6.

9) For the lattice structure, how many neighbors did individual spikes have in the data used to make the lattice reconstruction in Figure 4B?

In order to maximize the number of spikes that could be incorporated in our reconstruction, we included spikes with at least three neighbors, as determined by the PatchFinder script. This information is provided in the section entitled “Cryo-EM grid preparation, data acquisition and data processing” section of the Materials and methods.

Also, please comment on why the error bars are so large for the +4G2 spikes with 0, 1, or 2 neighbors in your chart in Figure 4B. This chart shows that there were examples in the data of spikes with less than 8 neighbors.

The large error bars (more accurately, the minimum and maximum values, as these are box and whisker plots) reflect the heterogeneity in lattice presentation on different VLPs that results from 4G2 treatment. Indeed, as can be observed in Figure 4—figure supplement 2, we observe that the Fab has a differential effect upon the spike presentation for the different VLPs tested. We can only speculate that this may be due to the intrinsic pleomorphic nature of hantaviral lattice assemblies, and the local environment of the spikes therein. As revealed by in Figure 4D, even in the absence of 4G2 we would expect there to be many spikes that have less than the full complement of neighbors (8) due to the pleomorphic nature of hantaviral virions.

To clarify this observation, we have added the following text to the Results section of the manuscript:

“The frequency of lattice-free spikes (with zero neighbors) was higher in the presence of Fab P-4G2, while the frequency of lattice-bound spikes (with four or more neighbors) was higher in the particles not treated with Fab P-4G2 (Figure 4C-D and Figure 4—figure supplement 2). We note that treatment with Fab P-4G2 has had a heterogeneous effect on the lattice of different VLPs as can be observed in Figure 4—figure supplement 2.”

These spikes were aligned to derive the map shown in Author response image 1.

**Author response image 1. respfig1:** A representative example of a hantaviral lattice “break point” displayed as a Mercator projection. Points colored blue represent points that were manually picked and aligned to generate the map in Author response image 2. This figure is adapted from Figure 4—figure supplement 2.

**Author response image 2. respfig2:** A reconstruction of a hantaviral spike using particles manually picked from “break points” in the lattice. The resulting map displays an empty patch of membrane where an 8^th^ neighbour would normally be expected as indicated with a dashed circle. Additional density that likely corresponds to fab P-4G2 is indicated with an asterisk. This reconstruction is generated from 160 manually picked and aligned particles and displayed at a resolution of 50 Å for clarity.

In the map shown in Author response image 2, it is possible to identify five neighboring spikes with two additional less well defined neighbors in the top right and left corners. These neighbors are less well defined as their position varies slightly depending on the geometry of the breakpoint. In the final position, where the eighth neighbor would be expected, there is a blank space of exposed membrane. Interestingly, we also see a small blob of density, which may correspond to the Fab 4G2 bound to an exposed Gc. While this reconstruction results in potentially interesting features, it arises from only 160 particles (and even fewer for all other datasets and types of lattice breaks) and is of relatively poor quality and resolution (22 Å). Given the limitations of this analysis, and that it is not central to our results or conclusions, we respectfully request not to include it in our revised manuscript.

Are the authors able to make reconstructions of spikes with less than 8 neighbors using subsets of the data? What would those look like?

In line with this comment, we have performed a reconstruction that specifically includes spikes with less than 8 neighbors. These spikes were manually picked from “break points” in the lattice:

How many surface proteins in their VLPs are bound with Fab?

From the spikes that were detected and used in our reconstructions, 1,721 (~46%) out of 3,719 4G2-treated VLP spikes were bound to the Fab. We have included this information in Table S2 within Supplementary File 1 of our revised manuscript.

How did the authors pick the concentrations for making their Fab/VLP samples?

The concentration of PUUV 4G2 Fab used (~2.7 μM, 0.12 mg/ml) for preparing our Fab−VLP complex was selected in attempt to as the highest concentrations possible without overcrowding the images acquired by cryo-EM. This concentration is higher than the upper plateau observed in the neutralization curve displayed in Figure 1B.

10) Overall, the paper is missing some numbers that are necessary sanity checks for structures as well as the proposed mechanisms.

We would like to thank for the opportunity to clarify why many of the metrics the reviewers would like to see in the paper have not been included. As we explain in more detail below, low signal-to-noise ratio in cryo-ET impedes counting the absolute number of spikes per VLP. For the same reason, we cannot quantify the unoccupied areas. Furthermore, measuring the membrane area itself is technically complicated as particles are often not spherical or even ellipsoidal but rather pleomorphic in their shape. Despite these limitations, we are confident however that it is accurate to compare relative differences in spike numbers (or number of neighbors) as we have done to propose the mechanism for how 4G2 sequesters spikes from the lattice. Please see below for answers to more detailed points.

a) How big are PUUV virions? How big are the VLPs?

To the best of our knowledge, the diameter of PUUV virions has not been reported. For hantaviruses in general, previous analyses have demonstrated that those virions that exhibit a spherical morphology have an average diameter of approximately 75–160 nm (Huiskonen et al., 2010; Martin et al., Arch. Virol., 1985; Parvate et al., Viruses, 2019). PUUV VLPs had a median diameter of ~92-nm, as observed in the frequency plot shown in Figure 4—figure supplement 1. Additionally, we have added scale-bars to each of the tomographic slices presented in Figure 4—figure supplement 2.

b) What is their total surface area of the VLPs?

As presented in the histogram plot added to Figure 4—figure supplement 1, the maximum dimension in our VLPs is approximately 50-150 nm (median approximately 95 nm). Assuming perfectly spherical shape of the VLPs, this would correspond to a range in surface area of approximately 7,900 nm^2^ to 71,000 nm^2^, however the particles are not always spherical. As the membrane is not visible in the tomographic reconstructions in the upper and lower Z-slices due to the missing wedge, accurate modelling of the entire surface to measure the total surface area is difficult.

How many surface particles are on the VLPs?

In general, the number of particles assigned as spikes was up to approximately 200 per VLP but this is an underestimate as the areas on the poles of the particle were excluded from the analysis. Furthermore, due to low signal-to-noise ratio, which is an inherent limitation of the cryo-ET method, it is not possible to report the absolute number of spikes on a given VLP with any reasonable level of confidence. We would like to stress that this is true for any cryo-EM particle picking exercise, whether 2D or 3D, where the goal is to “pick” the correct particles of interest, and the number of “picks” depends on the selection criteria used, such as the cross correlation coefficient cutoff used to label a “pick” as a true positive (use of less stringent criteria result in more “picks” and more false positives). Here, we are only able to report the number of particles that were assigned as spikes by template matching using the set of criteria provided in the Materials and methods, and then used in the reconstructions.

c) How much space is left on the surface of the virion for particles to dissociate? In other words, does the 3D space on the VLP surface make sense for the spreading out of the proteins with the Fabs bound? Figure 4—figure supplement 2 looks like the surface is very saturated on the VLP alone examples (left), especially in the tomographic slice.

We cannot reliably quantify the free space on the membrane surface, as we cannot quantify the absolute number of spikes accurately due to noise inherent in tomograms (see our response to 10b). Also measuring the accurate surface area of the VLPs is difficult due to noise and also the missing wedge, which is present in all tomograms generated from slab-shaped samples (any sample vitrified on a cryo-EM grid). It is clear however that the surface of the VLP cannot be fully saturated, as square-shaped spikes cannot geometrically cover a closed surface. This is visible in the tomographic slices that do reveal gaps between the spike lattice patches, as well based in our earlier work (Huiskonen et al., 2010). It is not clear how much, if at all, the spikes would need to spread out to accommodate the Fab though our interpretation is that these gaps create sufficient empty space to allow enough spreading of the Fab-bound spikes (which represent only a fraction of all spikes).

d) Why do the non-Fab VLPs look like they have more surface protein (Figure 4—figure supplement 2, left panels) than the Fab-bound VLPs (Figure 4—figure supplement 2, right panels)?

This impression may rise due to the difficulty in visualizing these surface proteins in tomographic slices and the differences in the appearance of an ordered spike lattices vs. free spikes. To better demonstrate that the coverage of surface protein between non-Fab VLPs and Fab-bound VLPs are broadly indistinguishable, we have improved the contrast of our tomographic slice representations in Figure 4—figure supplement 2, in our revised manuscript.

e) How big is the reconstructed patch of lattice vs. the total surface area available for the virus?

A reconstructed patch of lattice, which corresponds to a total of nine tetrameric spike complexes, would occupy approximately 1,500 nm^2^ of total surface area (the area occupied by one spike is ~163 nm^2^). This compares to an approximate total surface area range of 7,900 nm^2^ to 71,000 nm^2^ (as described in point 10b above). However this simple calculation assumes spherical shape which is not accurate.

What portion of the viral surface do they predict to be occupied by the surface proteins in a lattice organization?

As iterated above there is inherent uncertainty in giving absolute numbers due to low signal-to-noise ratio in the data. We maintain however our position that it is valid to compare the numbers of spikes between two samples (with and without 4G2) i.e. compare the relative trend in the numbers of neighboring spikes as we have done in Figure 4, as we have used the same criteria for both datasets, and as these datasets themselves have been acquired the same way.

11) It will be unclear to most readers what the "metastable homodimerization interface" is and how it fits into the virus lifecycle. This needs further context in the Introduction, or earlier in the Results (similar to the statement in the fourth paragraph of the Discussion). Related to this, the authors to clarify in the text that single spikes consist of (Gn-Gc)4 hetero-octamers. Related to point above, "Gc only in the monomeric state", and references to the monomeric state elsewhere in the text, are confusing because Gc is part of a heterooctameric Gn-Gc spike. Clarification is required to distinguish between Gc-dimerizing lattice interactions and intra-spike interactions.

We thank the reviewers for this opportunity to clarify the terminology used to describe the hantaviral Gn-Gc assembly, and have rephrased the instances of “monomeric Gc” as “lattice-free Gc”, which is defined in the text as “Gc molecules that are not integrated within the lattice assembly by Gc homo-dimer interactions.” This revision is included in the following, revised Results section:

“Previous studies have suggested that contacts between (Gn−Gc)_4_ spikes are mediated by Gc homo-dimers (Bignon et al., 2019; Huiskonen et al., 2010; Li et al., 2016), while lattice breaks are expected to display Gc (“lattice-free Gc”) molecules that are not integrated within the lattice assembly by Gc homo-dimer interactions.”

On a similar note, in the Discussion, the authors say "The identification of the Gc oligomerization-specific epitope…" This wording is potentially misleading. Calling it “the Gc…” makes it sound like this is the only possible epitope for an antibody that would disrupt oligomerization. This may not be true. In addition, the phrase "oligomerization-specific epitope" makes it sound like this is the endogenous interaction site for the oligomerization between trimers to form a lattice, which the authors don't have any proof for because they don't dock structures into the oligomer lattice map that would potentially show the interfaces between trimers, nor do they make any mutations to disrupt the interaction to show that the lattice goes away on a VLP without the antibody present. Rephrasing to something like “The identification of an epitope that disrupts trimer oligomerization…” might be more appropriate.

We thank the reviewers for this helpful suggestion. As suggested, we have clarified this text as follows:

“Characterization of the Fab P-4G2 epitope has important implications for rational therapeutic design efforts and indicates that a lattice-free conformation of the Gc can be targeted by the antibody-mediated immune response that arises during infection.”

Furthermore, as described above in response to comment 7, we have docked PUUV Gc and Gn structures to our lattice-incorporated PUUV VLP surface reconstruction (new Figure 6), and the implications with regards to the P-4G2 epitope are presented in new Figure 6—figure supplement 2.

12) There is a discrepancy in the phrasing between the Abstract and the main text as follows. In the Abstract, authors say, "analysis demonstrates that 4G2 binds to a multi-domain site on the PUUV Gc and precludes fusogenic rearrangements…" In the main text, the authors say "may sterically preclude the formation of a fusogenic configuration…" I find that their data does not concretely determine that the fab binding does preclude rearrangements, so the phrasing may preclude from their main text is most appropriate. The Abstract should be rewritten to match the phrasing in the main text.

In line with this comment, we have rephrased the Abstract as follows:

“Crystallographic analysis demonstrates that PUUV-4G2 binds to a multi-domain site on PUUV Gc and may preclude fusogenic rearrangements of the glycoprotein that are required for host-cell entry.”